



# Validation of temperature data from the RAman Lidar for Meteorological Observations (RALMO) at Payerne. An application to liquid cloud supersaturation.

Giovanni Martucci[1], Francisco Navas-Guzmán[1], Ludovic Renaud[1], Gonzague Romanens[1], S. Mahagammulla Gamage[2], Maxime Hervo[1], Pierre Jeannet[3], and Alexander Haefele[1,2]

[1]Federal Office of Meteorology and Climatology, MeteoSwiss, Payerne, Switzerland
[2]Department of Physics and Astronomy, The University of Western Ontario, London, Canada
[3]Federal Office of Meteorology and Climatology, MeteoSwiss, Payerne, Switzerland (retired)

**Correspondence:** Giovanni Martucci (giovanni.martucci@meteoswiss.ch)

**Abstract.** The RAman Lidar for Meteorological Observations (RALMO) is operated at the MeteoSwiss station of Payerne (Switzerland) and provides, amongst other products, continuous measurements of temperature since 2010. The temperature profiles are retrieved from the pure rotational Raman (PRR) signals detected around the 355-nm *Cabannes* line. The transmitter-receiver system of RALMO is described in detail and the reception and acquisition units of the PRR channels are thoroughly

characterized. The *FastCom* P7888 card used to acquire the PRR signal, the calculation of the dead-time and the desaturation procedure are also presented. The temperature profiles retrieved from RALMO data during the period going from July 2017 to the end of December 2018 have been validated against two reference operational radiosounding systems (ORS) co-located with RALMO, i.e. the Meteolabor SRS-C50 and the Vaisala RS41. These radiosondes have also been used to perform seven calibrations during the validation period. The maximum bias ($\Delta T_{max}$), mean bias ($\mu$) and mean standard deviation ($\sigma$) of

RALMO temperature $T_{\mathrm{ral}}$ with respect to the reference ORS $T_{\mathrm{ors}}$ are used to characterize the accuracy and precision of $T_{\mathrm{ral}}$ in the troposphere. The $\Delta T_{max}$, $\mu$ and $\sigma$ of the daytime differences $\Delta T = T_{\mathrm{ral}} - T_{\mathrm{ors}}$ in the lower troposphere are $0.28\,\mathrm{K}$, $0.02\pm 0.1\,\mathrm{K}$ and $0.62\pm 0.03\,\mathrm{K}$, respectively. The nighttime differences suffer a mean bias of $\mu = 0.05\pm 0.34\,\mathrm{K}$, a mean standard deviation $\sigma = 0.66\pm 0.06\,\mathrm{K}$, and a maximum bias $\Delta T_{max} = 0.29\,\mathrm{K}$ over the whole troposphere. The small $\Delta T_{max}$, $\mu$ and $\sigma$ values obtained for both daytime and nighttime comparisons indicate the high stability of RALMO that has been calibrated only

seven times over 18 months. The retrieval method can correct for the largest sources of correlated and uncorrelated errors, e.g. signal noise, dead-time of the acquisition system and solar background. Especially the solar radiation (scattered into the field of view from the Zenith angle $\Phi$) affects the quality of PRR signals and represents a source of systematic error for the retrieved temperature. An imperfect subtraction of the background from the daytime PRR profiles induces a bias of up to $2\,\mathrm{K}$ at all heights. An empirical correction $f(\Phi)$ ranging from 0.99 to 1, has therefore been applied to the mean background of the PRR

signals to remove the bias. The correction function $f(\Phi)$ has been validated against the numerical weather prediction model COSMO suggesting that $f(\Phi)$ does not introduce any additional source of systematic or random error to $T_{\mathrm{ral}}$. A seasonality study has been performed to help understanding if the overall daytime and nighttime zero-bias hides seasonal non-zero biases that cancel out when combined in the full dataset. Finally, the validated RALMO temperature has been used in combination



with the humidity profiles retrieved from RALMO to calculate the relative humidity and to perform a qualitative study of supersaturation occurring in liquid stratus clouds.

## 1 Introduction

Continuous measurements of tropospheric temperature are essential for numerous meteorological applications and in particular for numerical weather predictions, for satellite CAL/VAL applications (Stiller et al., 2012; Wing et al., 2018) and for the understanding of climate change. Co-located temperature and humidity measurements allow to calculate the relative humidity, a parameter playing a key role in several thermodynamic processes, such as the hygroscopic growth of condensation nuclei, fog and cloud formation. When considering the thermodynamic processes occurring within a stagnant air mass, a strong increase in relative humidity it is often a precursor of fog, while the onset of supersaturation is linked to a consolidated radiation fog or a cloud forming at the top of a convective layer. Another important thermodynamic parameter is the convective available potential energy (CAPE); the CAPE is directly related to the temperature difference between two layers in the atmosphere. The knowledge of temperature as a function of altitude allows to monitor the atmospheric thermodynamic stability and to diagnose and forecast the onset and intensity of a thunderstorm. Despite its importance in all these processes, the atmospheric temperature is still undersampled in the lower troposphere where the traditional and well established observing systems (e.g., radiosounding, AMDAR, Mode-S, satellites) do not provide continuous measurements. A vertical profile of temperature in the troposphere can be measured efficiently by ground-based remote sensing instrumentation; differently from other technologies, remote sensing is best suited to operate continuously and to satisfy real-time data delivery requirements. Moreover, remote sensing instruments operating continuously for many years ensure long time series of data, which are fundamental for climatology studies. This study focuses on the measurement of the atmospheric temperature done by a LIght Detection And Ranging (LIDAR) instrument. Best known methodologies to retrieve temperature profiles using a LIDAR can be split into four groups of techniques, the differential absorption LIDAR (DIAL), the high spectral resolution LIDAR (HSRL), the Rayleigh and the Raman techniques (Wulfmeyer et al. (2015) and references therein). Measurements with DIAL are based on the dependency of the molecular absorption on the atmospheric temperature, namely oxygen molecules with their constant mixing ratio in the dry atmosphere are used as targets by DIAL to retrieve the temperature profile (Behrendt, 2005; Hua et al., 2005). The HSRL technique uses the Doppler frequency shifts produced when photons are scattered from molecules in random thermal motion; the temperature dependence of the shape of the *Cabannes* line is used directly for temperature measurements (Theopold and Bösenberg, 1993; Wulfmeyer and Bösenberg, 1998; Bösenberg, 1998). The Rayleigh method is based on the assumption that measured photon-count profiles are proportional to the atmospheric mass-density profile in a atmosphere that behaves like an ideal gas and that is in hydrostatic equilibrium. The mass-density profile is used to determine the absolute temperature profile (Hauchecorne et al., 1991; Alpers et al., 2004; Argall, 2007). The Pure Rotational Raman (PRR) method relies on the dependence of the rotational spectrum on atmospheric temperature (Cooney, 1972; Vaughan et al., 1993; Balin et al., 2004; Behrendt et al., 2004; Di Girolamo et al., 2004; Achert et al., 2013; Zuev et al., 2017). A combination of the Rayleigh and Raman methods is also possible and allows to extend significantly the atmospheric region where the temperature is retrieved (Li et al.,



2016; Gerding et al., 2008). The four methods have the common objective to produce a temperature profile as close as possible to the true atmospheric status. In the attempt of doing that, a reference must be used to calibrate the LIDAR temperature and calculate the related uncertainty. Trustworthy references can be provided by co-located radiosondes, satellites or a numerical models. A co-located RS can act as reference to calibrate and monitor the stability of a LIDAR system over long periods of time (Newsom et al., 2013).Our study presents a characterization of the radiosounding systems (RS) in use at Payerne and their validation with respect to the Vaisala RS92 certified by the Global Climate Observing System (GCOS) Reference Upper-Air Network (GRUAN). Assimilation experiments using validated Raman LIDAR temperature profiles have been performed, among others, by Adam et al. (2016); Leuenberger et al. (2020). Both studies highlight the big potential of Raman LIDAR to improve numerical weather prediction (NWP) models through data assimilation (DA).

In this study we characterize and validate RALMO temperature profiles and demonstrate the high stability of the system. The paper is organized as follows: In Section 2 we establish the quality of the reference radiosonde data sets. The LIDAR system is described in detail in Section 3 followed by an uncertainty estimation in Section 4. In Sections 5 and 6 we present the statistics of the comparison between LIDAR and radiosondes.Moreover, the validated RALMO temperature has been used in combination with the humidity profiles retrieved from RALMO to calculate the relative humidity and to perform a qualitative study of supersaturation occurring in liquid stratus clouds (Section 7).

The maximum ($\Delta T_{max}$) and mean bias ($\mu$) of the difference ($\Delta T$) of several LIDAR temperature profiles with respect to temporally and spatially co-located radiosonde profiles represent the systematic uncertainty of the LIDAR temperature. The variability of all differences $\Delta T$ over the entire dataset yields the random uncertainty ($\sigma$) of the LIDAR temperature. In section 5 we present the statistical analysis of the $\Delta T = T_{\mathrm{ral}} - T_{\mathrm{ors}}$ dataset and we analyze the possibles causes of $\mu$ and $\sigma$ over the period July 2017–December 2018. An additional statistical study has been performed splitting the $\Delta T$ dataset into seasons to investigate the effect of solar background and its correction function $f(\Phi)$ on the retrieved temperature profiles in terms of $\mu$ and $\sigma$ (Section 6). Moreover, the validated RALMO temperature has been used in combination with the humidity profiles retrieved from RALMO to calculate the relative humidity and to perform a qualitative study of supersaturation occurring in liquid stratus clouds (Section 7).

## 2 Validation of the reference radiosounding systems

In the framework of the operational radiosonde flight programme, the operational radiosonde is launched twice daily at Payerne at 11 UTC and 23 UTC (in order to reach 100 hPa by 00 UTC and 12 UTC) and provides profiles of humidity ($q$), temperature ($T$), pressure ($P$) and wind ($u$). In addition to the operational programme, MeteoSwiss is part of the Global Climate Observing System (GCOS) Reference Upper-Air Network (GRUAN) since 2012 with the Vaisala sonde RS92. In the framework of GRUAN, MeteoSwiss has launched from the aerological station of Payerne more than 300 RS92 sondes between 2012 and 2019, contributing significantly to its characterization (metadata, correction algorithms and uncertainty calculation) and to its GRUAN certification (Dirksen et al., 2014; Bodeker and Kremser, 2015). Before being part of GRUAN and since 2005, MeteoSwiss has used the RS92 sonde as *working standard* in the framework of the quality assurance programme of the different





versions of the Meteolabor Swiss RadioSonde (SRS). Different versions of the SRS systems were operated at Payerne since 1990, starting from the analog SRS-400 (from 1990 to 2011) getting to the digital sondes SRS-C34 and C50. Starting from 2012, different versions of the SRS-C34 and SRS-C50 have been compared to the RS92 in the framework of GRUAN. In 2014, the Vaisala RS41 (Dirksen et al., 2019) was added to the GRUAN programme where it performed numerous multi-payload

flights with the RS92 and the SRS carried under the same balloon.

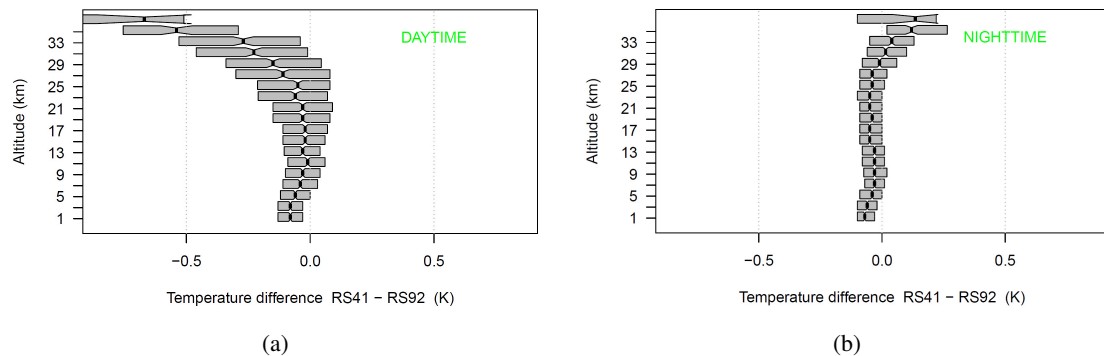

(a)                  (b)

**Figure 1.** Temperature deviation of the RS41 with respect to the RS92 calculated from the GRUAN multi-payload flights dataset during 2015/06 - 2018/12. The boxes are centered in the mean bias and span the $25^{th} - 75^{th}$ percentile range. Statistics are based on 58 flights at 11 UTC (Fig.1a) and 59 at 23 UTC (Fig.1b).

During the studied period, two different operational radiosounding systems (ORS) have been launched regularly at 11 UTC and 23 UTC, the SRS-C50 (February 2017- March 2018) and the Vaisala RS41 (March-December 2018 ). Thanks to the multi-sensor flights performed with the SRS-C50, the Vaisala RS41 and the Vaisala RS92, the SRS-C50 and RS41 have been validated by the GRUAN-certified RS92. Figures 1 and 2 show the statistical biases of the RS41 and the SRS-C50 with respect

to the reference RS92 as a function of height for the day and night-time launches. The differences have been co-added into altitude boxes of $2\,\mathrm{km}$ and the profiles have been sampled every $30\,\mathrm{s}$ starting from $15\,\mathrm{s}$ after launch. The boxes in the plots have boundaries at the $25^{th}$ and $75^{th}$ percentile and are centered (black dot in each box) in the mean value bias.

The RS41 and the SRS-C50 show an overall negative bias during both day and night never exceeding $-0.1\,\mathrm{K}$ along the whole troposphere. Only in the mid-stratosphere, above $30\,\mathrm{km}$, the daytime biases reach $-0.5\,\mathrm{K}$. In the framework of our

study, the region of interest for the temperature profiles measured by RALMO is the troposphere and, more rarely, the UTLS (upper troposphere and lower stratosphere, $\approx$ 0-14 km). In this region, as the statistic show, the two ORS perform very well. For the daytime comparisons (11 UTC), the mean bias of the RS41 over the region 0-14 km is $-0.05\,\mathrm{K} \pm 0.03\,\mathrm{K}$ with a mean standard deviation of $0.15\,\mathrm{K} \pm 0.05\,\mathrm{K}$. For the nighttime comparisons (23 UTC), the mean bias of the RS41 over the region 0-14 km is $-0.05\,\mathrm{K} \pm 0.02\,\mathrm{K}$ with a mean standard deviation of $0.11\,\mathrm{K} \pm 0.06\,\mathrm{K}$. The statistics of the SRS-C50 for daytime

(11 UTC) show a mean bias over the region 0-14 km of $-0.08\,\mathrm{K} \pm 0.02\,\mathrm{K}$ and a mean standard deviation of $0.19\,\mathrm{K} \pm 0.09\,\mathrm{K}$. For the nighttime comparisons (23 UTC), the mean bias of the SRS-C50 over the region 0-14 km is $-0.01\,\mathrm{K} \pm 0.02\,\mathrm{K}$ with a mean standard deviation of $0.13\,\mathrm{K} \pm 0.04\,\mathrm{K}$.





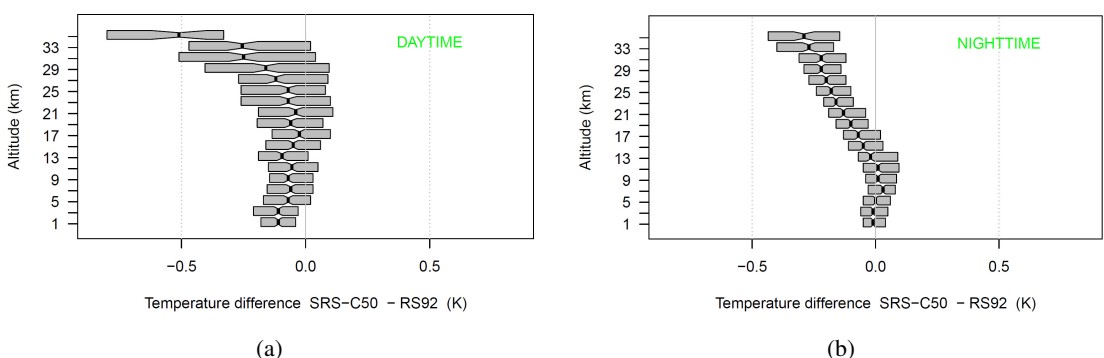

**Figure 2.** Temperature deviation of the SRS-C50 with respect to the RS92 calculated from the GRUAN multi-payload flights dataset during February–December, 2018. The boxes are centered in the mean bias and span the $25^{th} - 75^{th}$ percentile range. Statistics are based on 25 flights at 11 UTC (Fig.2a) and 26 at 23 UTC (Fig.2b.

The comparisons with the RS92 show that for both ORS the daytime differences undergo a larger variability along the 0-14 km vertical range compared to the nighttime statistics. The main reason for the larger variability is that, during the daytime flights, the RS92 and the two ORS undergo different exposure to the solar radiation, which causes a different response of the thermocouple sensors. The effect on the thermocouple becomes larger with the altitude as the solar radiation increases with height. All RS are corrected by the manufacturer for the effects of solar radiation on the thermocouple sensors. However, different manufacturers use different radiation corrections, which contributes to the statistical broadening of the differences at all levels. The overall (11 UTC and 23 UTC) performance of the two ORS in terms of bias with respect to the reference RS92 is summarized in figure 3. The distribution and mean value of the differences confirm that in the first $15\,\mathrm{km}$ the two ORS remain well below the $-0.1\,\mathrm{K}$-bias. The RS41 shows closer values to the RS92 than the SRS-C50 especially in the stratosphere. The *better* statistics of the RS41 should be interpreted also in light of the fact that the RS92 and the RS41 are both manufactured by Vaisala.

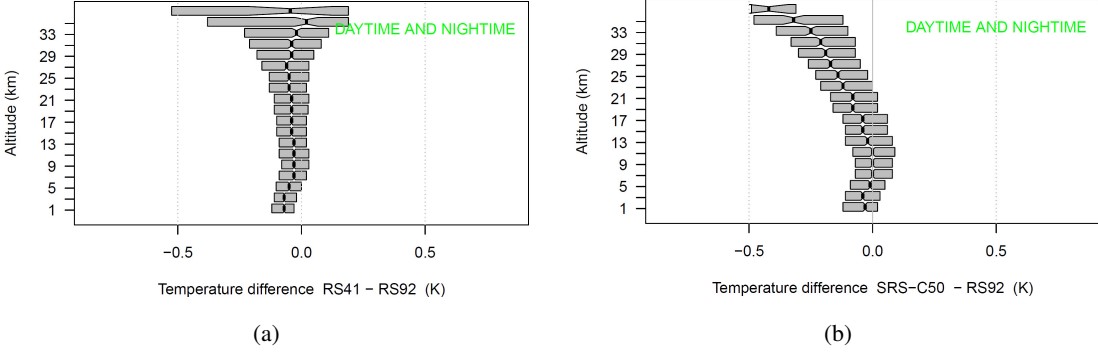

**Figure 3.** Overall temperature deviations at 11 and 23 UTC of the RS41 (Fig. 3a) and the SRS-C50 (Fig. 3b) with respect to the reference sonde RS92. The interpretation of the graphics is the same as for the previous figures



## 3 The RAman LIDAR for Meteorological Observations - RALMO

RALMO was designed and built by the École Polytechnique Fédérale de Lausanne (EPFL) in collaboration with MeteoSwiss. After its installation at the MeteoSwiss station of Payerne (N $46°48.0'$, E $6°56.0'$, $491$ m a.s.l.) in 2007 it has provided profiles of $q$, $T$ and aerosol backscatter ($\beta$) in the troposphere and lower stratosphere almost uninterruptedly since 2008 (Brocard et al.,
2013; Dinoev et al., 2013). The $T$ data during the 2008-2010 period are unexploited due to low quality of the analog channel. RALMO has been designed to achieve a measurement precision better than $10\,\%$ for $q$ and $0.5$ K for $T$ with a $30$ min integration time and to reach at least $5$ km during daytime and $7$ km during nighttime in clear-sky conditions. RALMO uses high-energy emission, narrow receiver's field of view and a narrow-band detection to achieve the required daytime performance. The data acquisition software has been developed to ensure autonomous system's operation and real-time data availability. RALMO's
tripled Nd:YAG laser emits $400$ mJ per pulse at $30$ Hz and at $355$ nm. A beam expander expands the beam's diameter to $14$ cm and reduces the beam divergence to $0.09$ mrad$\pm 0.02$ mrad. The returned signal is an envelope of the $355$ nm elastic- and Raman-backscattered signals, i.e., PRR, water vapour, oxygen, nitrogen and Rayleigh. Next, the Raman lidar equation (RLE) for the PRR signal is presented along with the detailed description of how RALMO selects the high and low quantum shift wavelengths used in the RLE to retrieve the temperature.

### 3.1 Pure Rotational Raman Temperature

Raman LIDAR measurements of the atmospheric temperature rely on the interaction between the probing electromagnetic signal at wavelength ($\lambda$) emitted by the LIDAR and the molecules of $O_2$ and $N_2$ encountered along the probing path. In addition to the Rayleigh light backscattered by the aerosols and molecules at the same frequency as the incident light, the $O_2$ and $N_2$ molecules return a frequency-shifted Raman signal back to the LIDAR's receiver. The Raman-backscattered signal
is shifted in frequency due to the rotational and vibrational Raman effect. In this study only the pure rotational part of the spectrum around the Rayleigh frequency (*Cabannes* line) is detected by RALMO and analyzed (Fig. 4).

The Raman LIDAR equation, RLE, yields the intensity of the PRR signal $S_{\mathrm{PRR}}$:

$$S_{\mathrm{PRR}}(z) = \frac{C}{z^2} O(z) n(z) \Gamma_{atm}^2(z) \left( \sum_{i=O_2, N_2} \sum_{J_i} \tau(J_i) \eta_i (\frac{d\sigma}{d\Omega})_{\Pi}^i (J_i) \right) + B \tag{1}$$

The received $S_{\mathrm{PRR}}$ signal measured over the time $t$ is a function of the altitude $z$; $C$ is the LIDAR constant; $O(z)$ is the
geometrical overlap between the emitted laser and the receiver's field of view; $n(z)$ is the number density of the air; $\Gamma_{atm}(z)$ is the atmospheric transmission; $\tau(J_i)$ is the transmission of the receiver for each PRR line $J_i$; $\eta_i$ is the volume mixing ratio of nitrogen and oxygen; $(\frac{d\sigma}{d\Omega})_{\Pi}^i (J_i)$ is the differential Raman cross section for each PRR line $J_i$ and $B$ is the background of the measured signal. Air mainly contains oxygen and nitrogen ($\approx 99\%$) whose ratio remains fairly constant in the first $80$ km of atmosphere, so $\eta_i$ can be regarded as a constant in eq. 1. The LIDAR constant $C$ depends on the overall efficiency of the
*transciever* (transmitter and receiver) system including the photo multipliers (PMT) efficiency, on the area of the telescope, and on the signal's intensity. The full expression of the differential Raman cross section for single lines of the PRR spectrum can be found in the reference book chapter by Behrendt (2005).





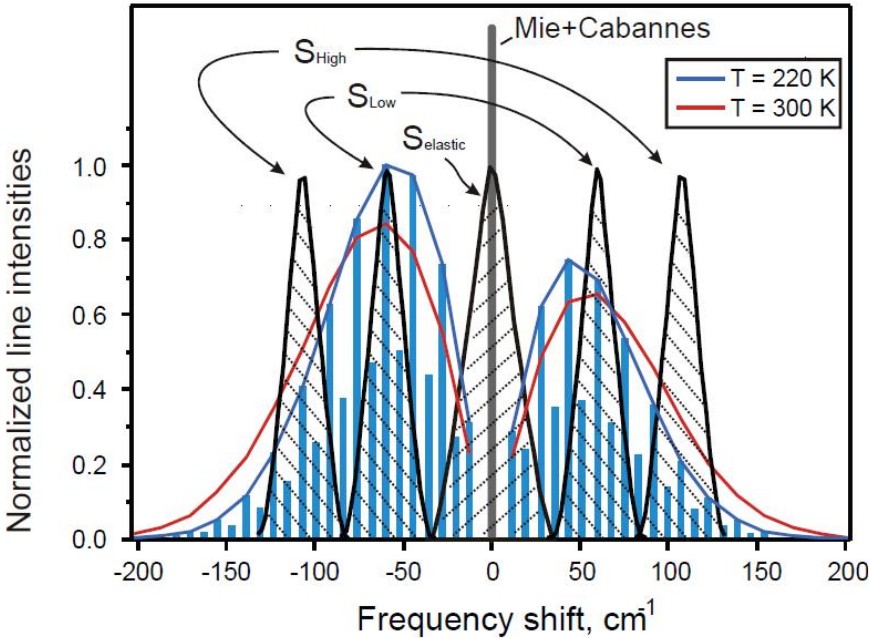

**Figure 4.** Temperature dependence of the *Stokes* and *Anti-Stokes* pure-rotational Raman spectrum of $N_2$ calculated at 220 K and 300 K. The intensity of the spectral lines are in normalized relative units. The wavelength scale is for a laser wavelength of 355 nm. The total intensity of the high and low quantum-shifted PRR channels is the summation of the single line intensities underneath the bell-shaped black curves.

## 3.2 Temperature polychromator of RALMO

The two-stage temperature polychromator, hereafter referred to as PRR polychromator, represents the core of the the signal selection. The PRR polychromator separates several pure-rotational Raman spectral lines and isolates elastic scattering consisting of Rayleigh and Mie lines (*Cabannes* line).

5 The PRR signal from the $O_2$ and $N_2$ atmospheric molecules is collected by four parabolic high-efficiency reflecting mirrors each one with diameter of 30 cm. The mirrors have dielectric reflection coating with $R > 99\%$ for the vibrational Raman wavelengths and $R > 96\%$ for the elastic and pure-rotational Raman for both cross and parallel polarized light. Nine-degrees tilted Semrock Razor Edge Filters (REF) are installed just below the focal points of two of the four mirrors (Fig. 5a). The REF are long-wavelength pass filters and have a cut-off wavelength at 364 nm (Fig. 5b). The ro-vibrational Raman scattering from the
10 atmospheric $H_2O$, $O_2$ and $N_2$ is transmitted by the REF onto the optic fibers placed above the REF at the exact focal distance of the parabolic mirrors. The elastic (Rayleigh and Mie) and PRR scattering are reflected by the tilted REF onto 0.4-mm optic fibers and transmitted to the PRR polychromator.

The two optic fibers transmitting the PRR and the elastic signals enter the temperature polychromator through the first fibers' block shown in Figure 6. The fibers are fixed into the fiber's block ensuring no or negligible temperature and mechanical-




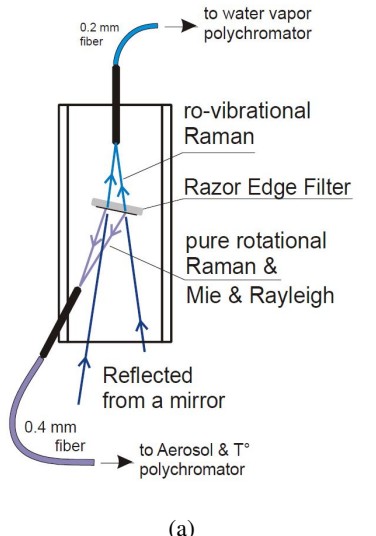

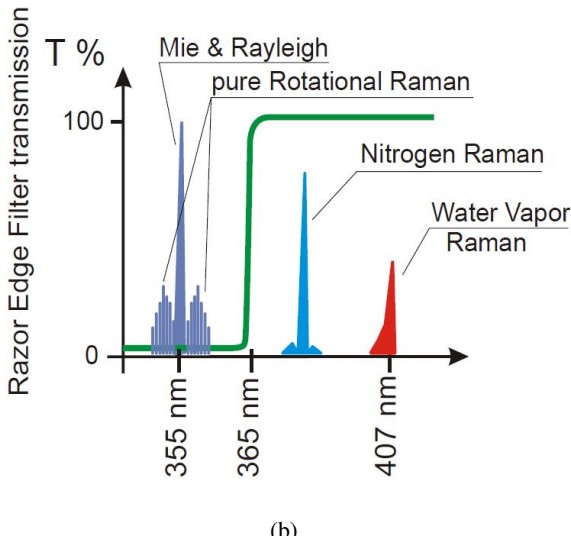

(a)                                                                      (b)

**Figure 5.** (a) Telescopic mirrors, Razor Edge Filter and optic fibers. (b) The REF cut-off frequency applied to the 355-nm Raman spectrum

induced drifts of the fiber alignment with respect to the other optical elements inside the polychromator (detailed in Fig. 7). The outline of the two fibers' blocks Cartesian coordinates system in figure 6a shows the position of the input, output and *intermediate* fibers (from stage−1 to stage−2) as a function of their *abscissa-ordinate*, $x - y$, positions. At $x = 21.5$ mm, the input fibers, coming from the mirrors, are located at the $y = 20$ mm and $y = 24$ mm, the output "elastic" fibers are located

at $y = 18$ mm and $y = 22$ mm. The two output elastic signals are then transmitted through the fibers and combined together just before entering the PMT installed outside the polychromator's box. The two input fibers transmit the PRR and the elastic signals onto an aspheric lens with focal length of 300 mm and diameter of 150 mm. The two signals are then transmitted through the lens onto a reflective holographic diffraction grating with groove density 600 grooves/mm oriented at a diffraction angle of $48.15°$ with respect to the axis of the lenses in a *Littrow* configuration. The two input signals (one from each mirror)

are diffracted by the grating polychromator and separated into high- and low quantum number lines from both *Stokes* and *Anti-Stokes* parts of the Raman-shifted spectrum. Two groups of four spectral lines are then diffracted, i.e. $J_{high}^{Stokes}$, $J_{high}^{AntiStokes}$, $J_{low}^{Stokes}$, $J_{low}^{AntiStokes}$.

The theoretical polychromator efficiencies $\xi$ ($\xi \in$ [0-1]) for the nitrogen and oxygen PRR lines $J_{high}^{Stokes}$, $J_{high}^{AntiStokes}$, $J_{low}^{Stokes}$ and $J_{low}^{AntiStokes}$ are shown in Table 1 and Table 2, respectively. The low and high quantum-number signals $J_{low}$ and $J_{high}$

Raman-backscattered by the nitrogen molecules are diffracted by the polychromator most efficiently at lines with quantum number $n = 6$ ($J_{low}^{AntiStokes}$, $\lambda = 353.97$ nm, $J_{low}^{Stokes}$, $\lambda = 355.47$ nm) and $n = 12$ ($J_{high}^{AntiStokes}$, $\lambda = 353.37$ nm, $J_{high}^{Stokes}$, $\lambda = 356.07$ nm). Similarly to the nitrogen, the PRR signals backscattered by the oxygen molecules are diffracted by the polychromator most efficiently at lines with quantum number $n = 9$ ($J_{low}^{AntiStokes}$, $\lambda = 353.96$ nm, $J_{low}^{Stokes}$, $\lambda = 355.48$ nm) and $n = 17$ ($J_{high}^{AntiStokes}$, $\lambda = 353.38$ nm, $J_{high}^{Stokes}$, $\lambda = 356.06$ nm).





**Table 1.** Theoretical polychromator efficiencies for the Nitrogen PRR lines

| Nitrogen | | | | | |
| --- | --- | --- | --- | --- | --- |
| $J_{low}$ | | | $J_{high}$ | | |
| $J_{low}^{AntiStokes}(n)$ | $\lambda$ | $\xi$ | $J_{high}^{AntiStokes}(n)$ | $\lambda$ | $\xi$ |
| 3 | 354.2501 | 0.0325 | 10 | 353.5532 | 0.2197 |
| 4 | 354.1503 | 0.2032 | 11 | 353.4540 | 0.6577 |
| 5 | 354.0505 | 0.6270 | 12 | 353.3549 | 0.9636 |
| 6 | 353.9509 | 0.9565 | 13 | 353.2559 | 0.6915 |
| 7 | 353.8513 | 0.7217 | 14 | 353.1570 | 0.2433 |
| 8 | 353.7519 | 0.2694 | 15 | 353.0582 | 0.0420 |
| $J_{low}$ | | | $J_{high}$ | | |
| $J_{low}^{Stokes}(n)$ | $\lambda$ | $\xi$ | $J_{high}^{Stokes}(n)$ | $\lambda$ | $\xi$ |
| 3 | 355.1511 | 0.0526 | 10 | 355.8543 | 0.2577 |
| 4 | 355.2514 | 0.2709 | 11 | 355.9548 | 0.6922 |
| 5 | 355.3518 | 0.7166 | 12 | 356.0554 | 0.9500 |
| 6 | 355.4523 | 0.9735 | 13 | 356.1560 | 0.6668 |
| 7 | 355.5527 | 0.6796 | 14 | 356.2566 | 0.2395 |
| 8 | 355.6532 | 0.2439 | 15 | 356.3572 | 0.0441 |

**Table 2.** Theoretical polychromator efficiencies for the Oxygen PRR lines

| Oxygen | | | | | |
| --- | --- | --- | --- | --- | --- |
| $J_{low}$ | | | $J_{high}$ | | |
| $J_{low}^{AntiStokes}(n)$ | $\lambda$ | $\xi$ | $J_{high}^{AntiStokes}(n)$ | $\lambda$ | $\xi$ |
| 5 | 354.2305 | 0.0493 | 15 | 353.5124 | 0.3759 |
| 7 | 354.0864 | 0.4535 | 17 | 353.3696 | 0.9524 |
| 9 | 353.9425 | 0.9598 | 19 | 353.2272 | 0.5496 |
| 11 | 353.7989 | 0.4686 | 21 | 353.0851 | 0.0727 |
| $J_{low}$ | | | $J_{high}$ | | |
| $J_{low}^{Stokes}(n)$ | $\lambda$ | $\xi$ | $J_{high}^{Stokes}(n)$ | $\lambda$ | $\xi$ |
| 5 | 355.1708 | 0.0765 | 15 | 355.8956 | 0.4195 |
| 7 | 355.3157 | 0.5454 | 17 | 356.0404 | 0.9457 |
| 9 | 355.4607 | | 19 | 356.1852 | 0.5308 |
| 11 | 355.6057 | | 21 | 356.3298 | 0.0747 |

Signals $J_{low}$ and $J_{high}$ are sums of the respective Stokes and anti-Stokes lines for nitrogen and oxygen. The eight $J-$signals diffracted by the polychromator are then re-focused by the aspheric lens onto the *intermediate* fibers positioned at the $y-$ordinates $y = 18$ mm and $y = 22$ mm and at the $x-$abscissae $x = 18.93$ mm, $x = 20.075$ mm, $x = 22.925$ mm, $x = 24.07$ mm.

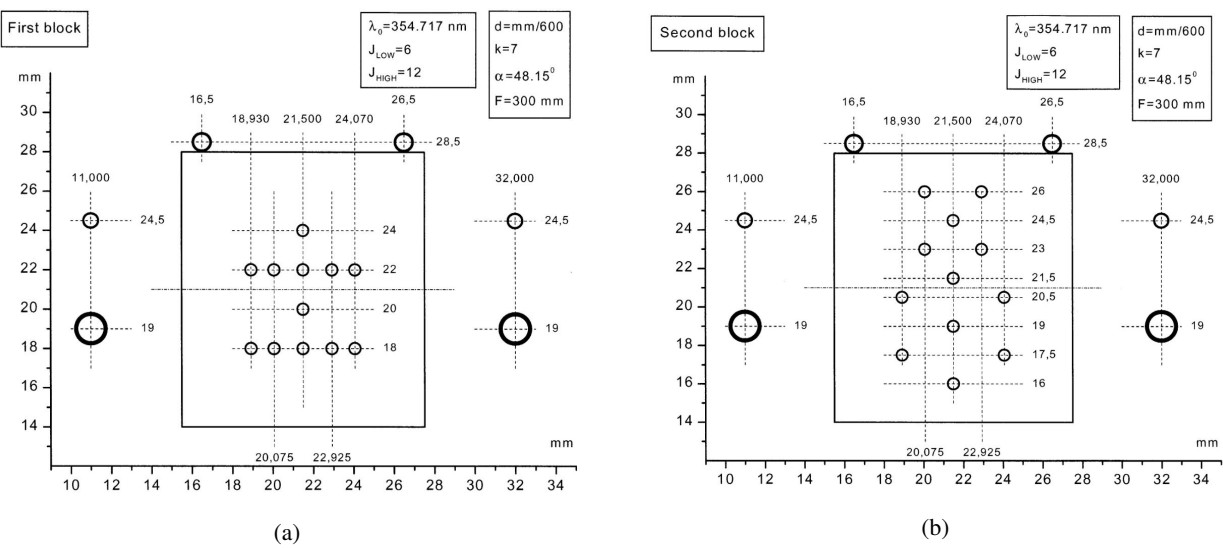

(a)
(b)

**Figure 6.** (a) Telescopic mirrors, Razor Edge Filter and optic fibers. (b) The REF cut-off frequency applied to the 355-nm Raman spectrum

The eight Stokes and Anti-Stokes $J-$signals are transmitted through the *intermediate* fibers into the second fibers' block (Fig. 6b) and subsequently transmitted along an optical path almost identical to the one in stage−1. Differently from stage−1, the eight $J-$signals are recombined by the diffraction grating polychromator into two groups of total $J-$signals ($J_{high}$ and $J_{low}$). The general outline of the the two-stage PRR polychromator is shown in Fig. 7.

The total $J-$signals are focused by the aspheric lens onto the output fibers positioned in the fibers' block in Fig. 6b at the same $x-$abscissa $x = 21.5$ mm and at the $y-$ordinates, $y = 24.5$ mm ($J_{high}$), $y = 21.5$ mm ($J_{high}$), $y = 19$ mm ($J_{low}$) and $y = 16$ mm ($J_{low}$). The output fibers transmit the four $J_{high}$ and $J_{low}$ signals from the two mirrors to two separate PMT boxes installed outside the polychromator unit. Inside each PMT box two $J-$signals are combined by an imaging system made by two lenses focusing onto a common spot. This last recombined signal is then divided by a beam splitter into two signals, one at 10 % and the other at 90 % of the intensity, which are focused onto two independent PMTs. A total of four signals are then obtained at the end of the receiver chain, i.e., $J_{high}^{10\%}$, $J_{high}^{90\%}$, $J_{low}^{10\%}$ and $J_{low}^{90\%}$

### 3.3 PRR channel acquisition system

The acquisition of RALMO's PRR channels have been migrated in August 2015 from the Licel acquisition system to the FAST ComTec P7888 (FastCom). The Model P7888 Series is one of the fastest commercially available multiple-event time digitizer with four inputs (one for each PRR channel) with very short acquisition system's dead-time and consequently minimum saturation effects of the photon-counting channels. Compared to the Licel acquisition system, FastCom acquires the PRR





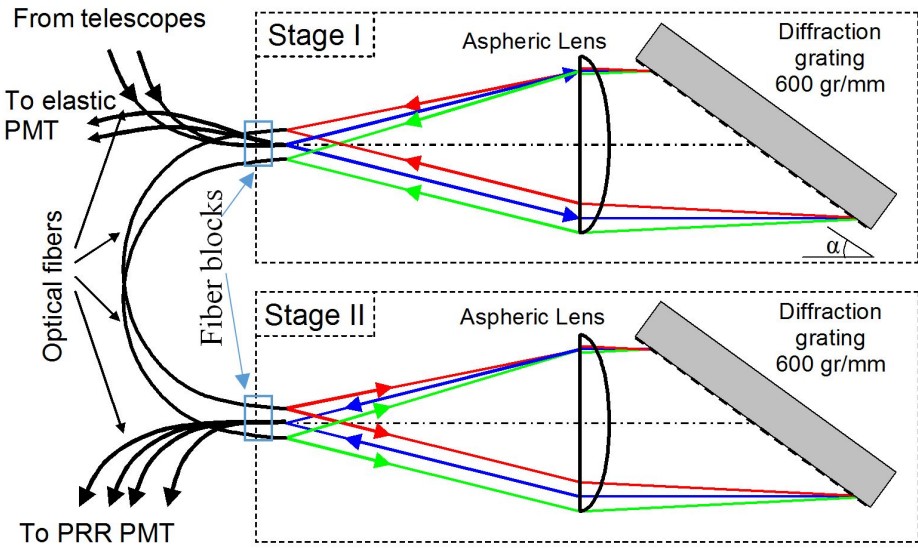

**Figure 7.** Optics block diagram of PRR polychromator

channels solely in photon counting mode, with higher range resolution and with about twice shorter dead-time, $\tau$. The FastCom acquisition system acquires two low-transmission channels ($J_{high}^{10\%}$, $J_{low}^{10\%}$) and two high-transmission channels, ($J_{high}^{90\%}$, $J_{low}^{90\%}$). Most photon-counting acquisition systems are limited in performance by the dead-time $\tau$, i.e. the minimum amount of time in which two input signals may be resolved as separate events. Whenever two consecutive photons impinge on the detector with separation time $t < \tau$ the system counts only one event. Certain type of acquisition systems can be corrected for the underestimation induced by $\tau$, the correction of the PRR signals measured by the FastCom system is presented in the section 4.

## 4   Retrieval of $T_{\mathrm{ral}}$ and calculation of the uncertainty

The high and low-frequency-shifted $S_{\mathrm{PRR}}$ signals have the expression given in eq.(1). In order to use them to retrieve the temperature profile, eq.(1) shall be corrected for the dead-time and the background. Once the signals are corrected, their ratio is used used to retrieve the temperature from eq.(3) scaled by two coefficients $A$ and $B$. The atmospheric temperature is then obtained from the calibration of eq.(3) with respect to $T_{\mathrm{ors}}$ and the determination of $A$ and $B$. The calibrated temperature is then provided along with its uncertainty. Table 3 summarizes the vertical and temporal resolution of the $S_{\mathrm{PRR}}$ signal at different stages of the data processing. The vertical resolution of $T_{\mathrm{ral}}$ is not constant with the altitude and depends on the calculated total random uncertainty in eq.(6). A Savitzy-Golay digital filter with polynomial degree $K = 1$ is applied to the $T_{\mathrm{ral}}$ profiles to degrade the sampling resolution and reduce the sampling noise. The adopted procedure and the definition of the obtained vertical resolution are compliant with the NDACC recommendations detailed in the work by Leblanc et al. (2016). The initial and highest resolution is $\delta z_{max} = 30\,\mathrm{m}$, which is degraded to a minimum $\delta z_{min} = 400\,\mathrm{m}$ corresponding to the regions where



the error is large (normally, the upper troposphere and lower stratosphere). For clear-sky measurement, an upper altitude cut-off is set at the altitude where the error exceeds $0.75\,\text{K}$, in presence of clouds, the upper limit is set by the cloud base detected by a colocated ceilometer. Very often, the clear-sky cut-off altitude corresponds to an altitude between 5 and 7 km during daytime measurements.

**Table 3.** Spatial and temporal resolution of $S_{\text{PRR}}$ signals and $T_{\text{ral}}$

| level | $\delta t$ in min | $\delta z_{min}$ in m | $\delta z_{max}$ in m |
|---|---|---|---|
| raw $S_{\text{PRR}}$ | 1 | 2.4 | 2.4 |
| corrected $S_{\text{PRR}}$ | 1 | 2.4 | 2.4 |
| $T_{\text{ral}}$ | 30 | 400 | 30 |

## 4.1 Correction of $S_{\text{PRR}}$

The PRR signals are corrected for the systematic underestimation of the true photon-counting signal (dead-time) and for the offsets (instrumental and solar). The first correction is then for the acquisition system's dead-time $\tau$. The low-transmission channels $J_{high}^{10\%}$ and $J_{low}^{10\%}$ do not become saturated and are used as reference channels to identify the saturation of the high-transmission channels $J_{high}^{90\%}$ and $J_{low}^{90\%}$. Assuming that the PMTs and the associated electronics obey the non-paralyzable assumption (Whiteman et al., 1992), we have studied the departure from the constant ratio $J^{10\%}/J^{90\%}$ as a function of $\tau$. We have applied a method based on the non-paralyzable condition (Newsom et al., 2009) to a year of data and have calculated $\tau$ for all the cases when the saturation clearly affected the high-transmission channels $J^{90\%}$.

As soon as the saturation has its onset, the ratio $J^{10\%}/J^{90\%}$ ceases to be constant and the saturated $J^{90\%}$ yields smaller count rates than the true ones. When the $J^{90\%}$ is desaturated using the correct $\tau$, it gives the $\tau-$corrected signal $J_{desat}(\tau) = J^{90\%}/(1 - \tau J^{90\%})$. One thousand linear fits $J^{10\%} = f(J_{desat}(\tau_i))$ are performed with $\tau_i$ varying in the interval $\tau_i \in [0\,\text{ns-}10\,\text{ns}]$ at steps of 0.1 ns. The linear fits $J^{10\%} = f(J_{desat}(\tau_i))$ are performed over a temporal interval of 30 min and a vertical range defined by the count rates range $0.5\,\text{MHz}$ to $50\,\text{MHz}$, respectively $C_{min}$ and $C_{max}$ in Fig.8. For each linear fit we calculate the value $e(\tau_i)$ that provides the *distance* in the count-rate domain between $J^{10\%}$ and $f(J_{desat}(\tau_i))$ as a function of $\tau_i$. The minimization of $e(\tau)$ with respect to $\tau_i$ determines the value of the acquisition system's dead-time $\tau_i = \tau_{min} \in [0\,\text{ns-}10\,\text{ns}]$ for each channel. The obtained value $\tau_{min}$ is used to desaturate $J^{90\%}$ and to reestablish the constant ratio $J^{10\%}/f(J_{desat}(\tau_{min})) = constant$. Figure 8 shows an example of calculation of $\tau_{min}$ for the high-transmission channel of $J_{low}$. The curve function $e(\tau)$ in the figure's left panel has a minimum at $\tau_{min} = 3\,\text{ns}$. On the right panel, the uncorrected and the $\tau-$corrected relations are shown. The uncorrected relation $J^{10\%} = f(J_{desat}(\tau = 0))$ (solid black) departs from the linear relation $J^{10\%} = f(J_{desat}(\tau_{min}))$ (dashed green) as soon as the count rates exceed the lower bound $C_{min}$ (dashed red). Applying this method to a year of data and collecting more than hundred cases we have determined the mean dead-times $\tau = 1.4\,\text{ns}$ and $\tau = 3\,\text{ns}$ for $J_{high}$ and $J_{low}$, respectively. The desaturated $J_{high}$ and $J_{low}$ are further corrected for the back-





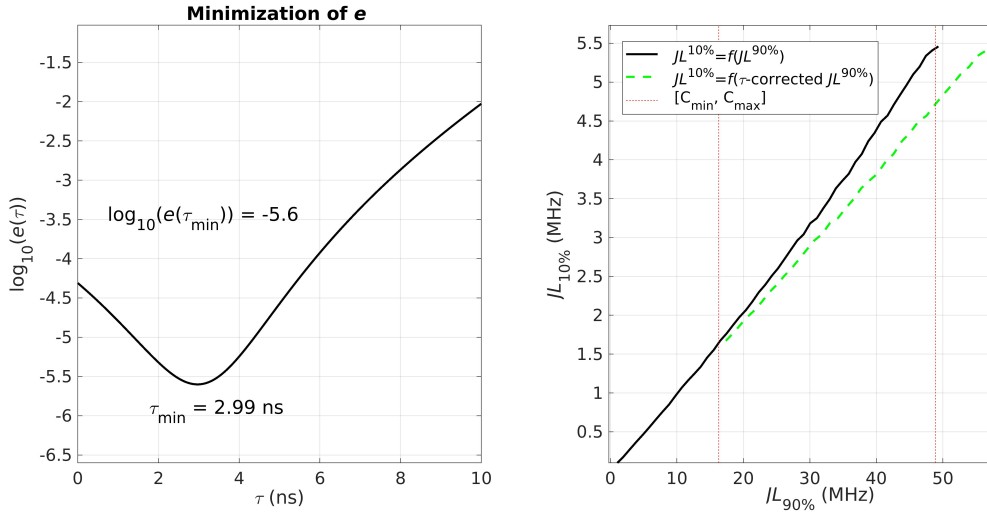

**Figure 8.** Dead-time calculation and correction of the high-transmission channel $J^{90\%}$. The left panel shows the minimization of the *distance* vector $e(\tau_i)$ in terms of $\tau_i$ yielding the deadtime $\tau = 2.99$ ns. The right panel shows the saturated $J^{10\%} = f(J_{desat}(\tau = 0))$ (solid black), the desaturated linear relation $J^{10\%} = f(J_{desat}(\tau_{min}))$ (dashed green) and the count-rate domain $[C_{min}, C_{max}]$ (dashed red)

ground and the procedure is described hereafter.

The electronic and solar background must be subtracted from $S_{\text{PRR}}$ before retrieving $T_{\text{ral}}$. While the electronic background is stable and does not undergo daily or seasonal cycles, the solar background changes in intensity with the position of the sun $\Phi$ (the angle between the zenith and the centre of the Sun's disc). We have found that subtracting the mean value of the far-range

signal ($z \in [50\text{-}60]$ km) from $S_{\text{PRR}}$, (subtraction of term $B$ from eq. 1) causes a systematic negative bias with respect to $T_{\text{ors}}$ of about 1 K at all altitudes $z$ during daytime. A relative change of 1% in the ratio $J_{low}/J_{high}$ due to an imperfect background subtraction can lead to a variation of up to 2 K in the retrieved temperature $T_{\text{ral}}$. Because the solar background ($SB$) dominates the total background of $S_{\text{PRR}}$, we focus on the correction of the background $B$ only as a function of the position of the sun. We have developed an empirical correction function $f(\Phi)$ applied to the background prior to subtraction from $S_{\text{PRR}}$. The function

$f(\Phi)$ is applied to the background $B$, and provides the corrected background $B_{corr} = f(\Phi) \cdot B$. Through the year's cycle, $B$ is reduced by a maximum amount of 1 % via the action of $f(\Phi)$. As eq. (2) shows, $f(\Phi)$ reaches daily minima when $\Phi = \Phi_{min}^{day}$ (noon), and returns to 100 % when $\Phi \geq 90°$ (after sunset and before sunrise). During the daily and annual cycle, $f(\Phi)$ then oscillates within the range $f(\Phi) \in [99\% - 100\%]$ reducing $B$ by the maximum amount of 1 % ($f(\Phi) = 99\%$) at noon on the $21^{st}$ of June when $\Phi = \Phi_{min}^{year}$.

$$f(\Phi) = 1 - 0.01 \cdot \frac{\cos \Phi}{\cos \Phi_{min}} \delta_\Phi, \quad \delta_\Phi \equiv 1 \quad \text{for} \quad 0 \leq \Phi < \pi/2, \quad \delta_\Phi \equiv 0 \quad \text{for} \quad \pi/2 \leq \Phi < 2\pi \tag{2}$$

If uncorrected, the retrieved daytime $T_{\text{ral}}$ suffers a bias at all heights with respect to $T_{\text{ors}}$. The bias is largest when $\Phi = \Phi_{min}^{day}$. The correction $f(\Phi)$ is applied only to the background of $J_{high}$. The intensity of $J_{high}$ is generally lower than $J_{low}$ at all





atmospheric temperatures (see Sect. 4), and so is also its signal-to-noise ratio (SNR). Even a small error of $\approx 1\%$ when subtracting $B$ from $J_{high}$ has a major impact on its SNR; $f(\Phi)$ corrects the imperfect subtraction of $B$ from $J_{high}$ and minimizes the daytime bias of $T_{\mathrm{ral}}$ with respect to $T_{\mathrm{ors}}$ almost perfectly.

## 4.2 Estimation of total random uncertainty

Once $B_{corr}$ is subtracted from the $\tau-$corrected $S_{\mathrm{PRR}}$, the deviation of $T_{\mathrm{ral}}$ from $T_{\mathrm{ors}}$ depends only on how precisely eq. (3), derived from the RLE, represents the true atmospheric temperature at the altitude $z$ and time $t$. The random uncertainty does not account then for the error induced by the saturation and the background, which are considered as purely systematic. The high-frequency-shifted, $J_{high}$, and low-frequency-shifted, $J_{low}$, signals in the Stokes and anti-stokes $Q-$branches depend on the temperature of the probed atmospheric volume (Fig. 4). The ratio of the $S_{\mathrm{PRR}}$ intensities $Q(z) = J_{low}(z)/J_{high}(z)$ is a

function of the atmospheric temperature $T$ at the distance $z$. Based on the calculations shown by Behrendt (2005) and for systems that can detect independent $J-$lines in each channel, the relationship between $T$ and $Q$ would take the form of eq. (3) with the *equals* sign. The approximation sign in eq. (3) indicates that the detection system detects more than one $J-$line and thus brings an inherent error. The calibration coefficients $A$ and $B$ are *a-priori* undetermined and can be determined by calibration of $T_{\mathrm{ral}}$ with respect to $T_{\mathrm{ors}}$.

$$T_{\mathrm{ral}} \approx \frac{A}{B + \ln Q}. \tag{3}$$

The coefficients $A$ and $B$ are determined calibrating $T_{\mathrm{ral}}$ with respect to $T_{\mathrm{ors}}$. The coefficient $A$ has units in Kelvin as eq. (3) is not normalized for the standard atmospheric temperature (Behrendt and Reichardt, 2000). The linear relation $y = A/(x+B)$ is used, where $x$ is the ratio $Q$ and $y$ is the reference temperature $T_{\mathrm{ors}}$. The mean error on $T_{\mathrm{ors}}$ for both the RS41 and SRS-C50 is $\approx -0.04 \pm 0.15$ K at 11 UTC and 23 UTC between 0 and $14\,\mathrm{km}$ (section 2). Due to the very small error on $T_{\mathrm{ors}}$, we can

calculate the uncertainty $U_{fit}$ of the fitting model only in terms of the fitting parameters' errors $\sigma_A$ and $\sigma_B$ (eq. 4). As it will be shown in the next section, the covariance $\sigma_{AB}$ of $A$ and $B$ is very close to zero ($< 10^{-3}$), thus $\sigma_A$ and $\sigma_B$ can be treated as statistically independent and used to calculate $U_{fit}$ from the first-order Taylor's series of propagation of fitting parameters' uncertainties.

$$U_{fit} = \frac{1}{(B + \ln Q)} \left[ \sigma_A^2 + \frac{A^2 \sigma_B^2}{(B + \ln Q)^2} \right]^{\frac{1}{2}}, \tag{4}$$

$U_{fit}$ is not the only error source, a second contribution to the total uncertainty comes from the fact that $S_{\mathrm{PRR}}$ is acquired by a photon-counting system and is affected by the measurement's noise that can be calculated using standard *Poisson* statistics. The error $\sigma_J$ for *Poisson*-distributed data is equal to the square root of the $S_{\mathrm{PRR}}$ signals, $\sigma_{J_{low}} = \sqrt{J_{low}}$ and $\sigma_{J_{high}} = \sqrt{J_{high}}$. In eq. (5), coefficients $A$ and $B$ can be regarded as independent from the noise on $S_{\mathrm{PRR}}$, as the contribution of it is already included in $\sigma_A$ and $\sigma_B$ in eq.(4).

$$U_{sig} = \frac{A}{(B + \ln Q)^2} \left[ \frac{1}{J_{high}} + \frac{1}{J_{low}} \right]^{\frac{1}{2}}. \tag{5}$$





The total uncertainty of the calibrated $T_{\mathrm{ral}}$ is a *type B* uncertainty (for Guides in Metrology, 2008) and is the sum of the independent error contributions $U_{fit}$ and $U_{sig}$.

$$U_T = \sqrt{U_{fit}^2 + U_{sig}^2} = \sqrt{\frac{1}{(B + \ln Q)^2}\left[\sigma_A^2 (B + \ln Q)^2 + A^2\left(\sigma_B^2 + \frac{1}{J_{high}} + \frac{1}{J_{low}}\right)\right]}. \tag{6}$$

### 4.3 Calibration of $S_{\mathrm{PRR}}$

5 For a very stable system like RALMO, calibrations can be performed once every few months to compensate for any occurring drift of the detection system's sensitivity and/or efficiency. Calibrations of RALMO are performed using eq. (3) in clear-sky conditions during nighttime to remove the effect of solar background and have a larger vertical portion of $T_{\mathrm{ral}}$ available for calibration (the daytime profiles have normally a lower cut-off altitude). Figure 9 shows a case of RALMO calibration, the green-shaded area represents $\pm 2U_T$ $(k = 2)$.

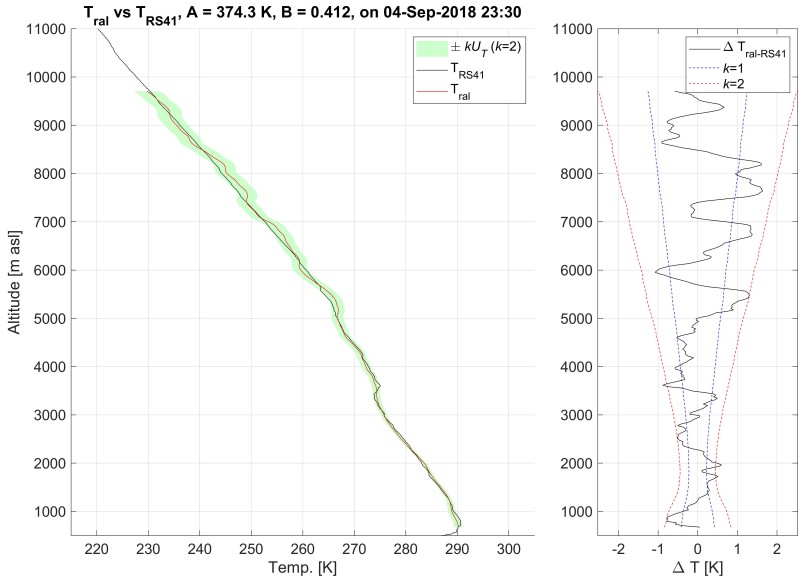

**Figure 9.** RALMO temperature profile calibrated by the ORS RS41. The left panel shows $T_{RS41}$ (solid black) and $T_{ral}$ (solid red) with the confidence interval (green shading) corresponding to $kU_T(k = 2)$. The right panel shows the differences $\Delta T_{ral-RS41}$ within the $U_T$ $k-$boundaries for $k = 1$ (dashed blue) and $k = 2$ (dashed red).

10 The calibrated $T_{\mathrm{ral}}$ results from the integration of 30 $\tau-$ and $B-$corrected $S_{\mathrm{PRR}}$ profiles ($\delta t = 30\,\mathrm{min}$) into eq. (3). At a given atmospheric altitude $z$ during the time interval $\delta t$, $U_T(z)|_{\delta t}$ is made of the single contributions $U_{sig}(z,t)$ and $U_{fit}(z,t)$. $U_{sig}(z,t)$ can be regarded as independent with respect to time; on the other hand, the errors $U_{fit}(z,t)$, depend on the atmospheric processes occurring within the layer $[z - \delta z/2, z + \delta z/2]$ during the time interval $[t - \delta t/2, t + \delta t/2]$ and are, *a-priori*, not statistically independent. By assuming that all errors in eq. (6) are statistically independent, we assume that the off-diagonal





elements of the *variance-covariance* matrix are all zero. By doing so, $U_T(z)|_{\delta t}$ could be underestimated by an amount equal to the non-zero covariance terms, including the covariance $\sigma_{AB}$. A method to assess the exhaustiveness of the theoretical error $U_T$ is to calculate how many points in the vector $\Delta T = T_{\text{ral}} - T_{\text{ors}}$ fall within the interval $[-k\sigma, +k\sigma]$ and check if they are compatible with the Gaussian probability levels 68.3%, 95.5% and 99.7% for $k = 1, 2, 3$, respectively. As it is shown in the

right panel of figure 9, almost all points along the vector $T_{\text{ral}} - T_{\text{ors}}$ fall within the interval $[-2\sigma, +2\sigma]$, i.e. the 98.1%. For $k = 1$ the percentage falls slightly below the expected level for a Normal distribution with only 61.2% of the points within $[-\sigma, +\sigma]$. Between July 2017 and December 2018, a total of seven calibrations have been performed (three SRS-C50 and four Vaisala RS41). The mean percentage of points over all performed calibrations is 65.1% for $k = 1$, 97.9.1% for $k = 2$ and 99.98% for $k = 3$. These values seem to confirm an overall exhaustiveness of $U_T$ with a slight underestimation of 3.2% at $k = 1$. The list

of calibrations is shown in Table 4. For each calibration, the table lists the date and end time of the calibration, the calibration coefficients $A$ and $B$ used in the fitting model eq. (3), the errors $\sigma_A$ and $\sigma_B$, the covariance $\sigma_{AB}$ and the ORS used to calibrate $T_{\text{ral}}$. As to further support the assumption of zero covariance of the coefficients $A$ and $B$, all covariances $\sigma_{AB}$ in the table are smaller than $1.71 \times 10^{-3}$. Between two consecutive calibrations performed at times $t_i$ and $t_{i+1}$ the coefficients $A(t_i)$ and $B(t_i)$ are used to calibrate all profiles $T_{\text{ral}}$ during the time interval $t \in [t_i, t_{i+1}]$.

**Table 4.** Calibrations of RALMO $T$ profiles by the *working standards*

| YYYY-MM-DD HH:MM | $A$ [K] | $B$ | $\sigma_A$ [K] | $\sigma_B$ | $\sigma_{AB}$ | ORS |
|---|---|---|---|---|---|---|
| 2017-07-06 23:30 | 372.97 | 0.42 | 0.7275 | 0.0027 | $0.78 \times 10^{-3}$ | SRS-C50 |
| 2017-08-24 23:30 | 375.63 | 0.43 | 1.1184 | 0.0041 | $1.71 \times 10^{-3}$ | SRS-C50 |
| 2017-10-16 23:30 | 376.44 | 0.42 | 0.5987 | 0.0022 | $0.93 \times 10^{-3}$ | SRS-C50 |
| 2018-04-21 23:30 | 372.94 | 0.41 | 0.6980 | 0.0026 | $0.76 \times 10^{-3}$ | RS41 |
| 2018-05-11 23:30 | 373.14 | 0.41 | 0.9510 | 0.0036 | $1.35 \times 10^{-3}$ | RS41 |
| 2018-07-07 23:30 | 374.42 | 0.41 | 0.9877 | 0.0037 | $1.46 \times 10^{-3}$ | RS41 |
| 2018-09-11 23:30 | 374.78 | 0.41 | 0.9276 | 0.0034 | $1.30 \times 10^{-3}$ | RS41 |

## 5   Validation of PRR temperature

More than 450 profiles $T_{\text{ral}}$ (245 nighttime, 215 daytime) have been compared to $T_{\text{ors}}$ and assessed separately for daytime and nighttime based on the bias and standard deviation ($\sigma$) of the differences $\Delta T = T_{\text{ral}} - T_{\text{ors}}$ over the period $1^{st}$ of July 2017- $31^{st}$ of December 2018. Two criteria to select the cases for the dataset have been used:

1. Only cases with no precipitation and no low clouds or fog are retained.

2. Only cases with $\Delta T < 5\,\text{K}$ are retained.



Criterion 1 is performed setting a threshold for minimum cloud base, $h_b$, at $1000\,\mathrm{m}$ (a.g.l.), for any values of $h_b < 1000\,\mathrm{m}$ $T_\mathrm{ral}$ is not retrieved. Whenever $h_b > 1000\,\mathrm{m}$, the cut-off altitude will correspond to $h_b$. Indeed, above $h_b$, the SNR drops abruptly and $U_T(z) \gg 1\,\mathrm{K}$. For this reason, especially during winter, when long-lasting stratus clouds occur in the altitude range $1\,\mathrm{km}$ to $4\,\mathrm{km}$, the nighttime and daytime $T_\mathrm{ral}$ is limited in range to the altitude $h_b$ (or are is calculated if $h_b < 1000\,\mathrm{m}$).

Criterion 2 is performed setting a threshold at $33\%$ of the number of elements along the profile $\Delta T$ exceeding $5\,\mathrm{K}$. If more than $33\%$ of the elements along $\Delta T$ exceed the threshold, the whole $T_\mathrm{ral}$ is rejected and not included in the statistics. For any value below the threshold, the outliers are removed from $T_\mathrm{ral}$. This is justified by the fact that exceedances counting more than $33\%$ are caused by temporary misalignment of the transceiver unit. On the other hand, exceedances well below the threshold can always occur (especially in the higher part of the profile) due to low SNR or unfiltered clouds. In Table 5 we present a

summary of the statistical parameters characterizing the daytime and nighttime differences $\Delta T$ that will be discussed in detail in the following sections. The dataset $\Delta T$ is described in terms of maximum mean bias $\Delta T_{max}$, average mean bias $\mu$, standard deviation $\sigma$ and maximum availability $N_{max}$ of the differences $\Delta T$ along the atmospheric range.

**Table 5.** Summary of the statistical parameters of the day and night $\Delta T$ dataset

| | | night | | |
|---|---|---|---|---|
| $\Delta T_{max}$ | $\mu$ | $\sigma$ | $N_{max}$ | range |
| $0.24\,\mathrm{K}$ | $0.05\pm 0.34\,\mathrm{K}$ | $0.66\pm 0.06\,\mathrm{K}$ | $244$ | $0.5{-}10\,\mathrm{km}$ |
| | | day | | |
| $\Delta T_{max}$ | $\mu$ | $\sigma$ | $N_{max}$ | range |
| $0.25\,\mathrm{K}$ | $0.02\pm 0.1\,\mathrm{K}$ | $0.62\pm 0.03\,\mathrm{K}$ | $212$ | $0.5{-}6\,\mathrm{km}$ |

Besides a global validation we also present and discuss the seasonal statistics in order to better characterize the system performance.

## 15  5.1  Nighttime temperature statistics

The nighttime $\Delta T_{max}$, $\mu$, $\sigma$ and $N_{max}$ of $\Delta T$ are are the metric to assess the accuracy and precision of $T_\mathrm{ral}$ with respect to $T_\mathrm{ors}$. Figure 10 shows $\Delta T_{max} = 0.24\,\mathrm{K}$, $\mu = 0.05\pm 0.34\,\mathrm{K}$, $\sigma = 0.66\pm 0.06\,\mathrm{K}$ and $N_{max} = 244$ over the tropospheric region $0.5\,\mathrm{km}$ to $10\,\mathrm{km}$.

In addition to the uncertainty assessment performed in section 4.3, the exhaustiveness of the theoretical total uncertainty

$U_T$ can be further assessed comparing $U_T$ with $\sigma$. The mean value of $U_T$ along the troposphere and over the seven nighttime calibrations is $\overline{U}_T = 0.64\,\mathrm{K}$, the mean nighttime standard deviation averaged over the tropospheric column in figure 10 is $\overline{\sigma} = 0.66\,\mathrm{K}$. The two $1-k$ uncertainties are then fully compatible.

The nighttime $\Delta T$ data are characterized by values of $\mu$ and $\sigma$ smaller than $1\,\mathrm{K}$, with minimum values in the lower troposphere from $0\,\mathrm{km}$ to $5\,\mathrm{km}$. It is indeed in the lower troposphere, where $\sigma$ is $\approx 0.6\,\mathrm{K}$, i.e. $0.1\,\mathrm{K}$ larger than the $0.5\,\mathrm{K}$ requirement





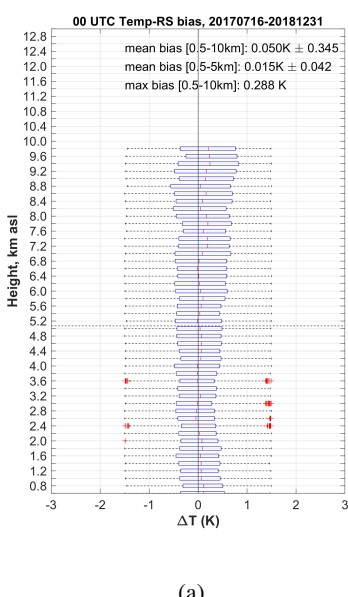

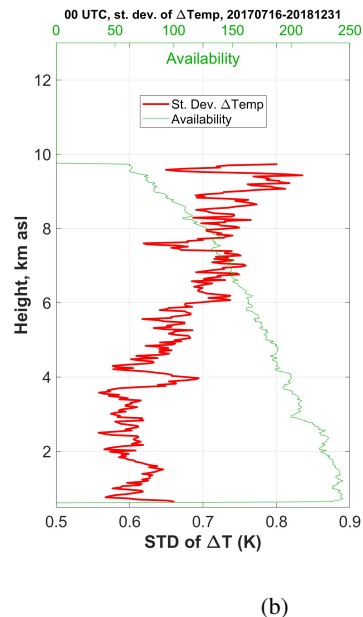

(a)  (b)

**Figure 10.** Nighttime bias and STD of $\Delta T$ over the period July 2017-December 2018. In fig.10a, on each box of $200\,\mathrm{m}$ vertical span, the central mark indicates the median, and the left and right edges of the box indicate the $25^{th}$ and $75^{th}$ percentiles, respectively. The whiskers extend to the most extreme data points not considered outliers, and the outliers are plotted individually and shown by the '+' symbol. Fig.10b shows the vertical profile of standard deviation (thick red) calculated over the altitude-decreasing number of $\Delta T$ points (thin green).

for data assimilation into the numerical weather prediction COSMO forecasting system (Fuhrer et al., 2018; Klasa et al., 2018, 2019). In order to achieve a successful assimilation of $T_{\mathrm{ral}}$ into COSMO, the overall impact of the assimilation shall correspond to an improvement of the forecasts without increasing the forecasts' uncertainty. Assimilation of high-SNR, well-calibrated $T_{\mathrm{ral}}$ into numerical models leads to the improvement of the forecasts. In the study by Leuenberger et al. (2020), the
5  authors assimilate, amongst other data, temperature and humidity profiles from RALMO showing the beneficial impact on the precipitation forecast over a wide geographical area.

## 5.2 Daytime temperature statistics

During daytime, the retrieved temperature profiles are limited in range to about $6\,\mathrm{km}$. Figure 11a and 11b show the bias $\Delta T$ and the standard deviation $\sigma$, respectively. The $\Delta T_{max}$, $\mu$, $\sigma$ and $N_{max}$ of the daytime $\Delta T$ over the lower troposphere ($0.5-6\,\mathrm{km}$)
10  are $0.25\,\mathrm{K}$, $0.02\pm0.1\mathrm{K}$, $0.62\pm0.03\mathrm{K}$ and 212, respectively. The data availability goes rapidly to zero above $5\,\mathrm{km}$.

The daytime $T_{\mathrm{ral}}$ profiles have been corrected for the solar background by $f(\Phi)$, which proves to be very efficient in removing the noon bias with respect to $T_{\mathrm{ors}}$. To ensure that the correction $f(\Phi)$ does not introduce any additional bias during the daily cycle, we have compared $T_{\mathrm{ral}}$ with the temperature calculated by the COSMO model. More than three months of clear-sky 24-





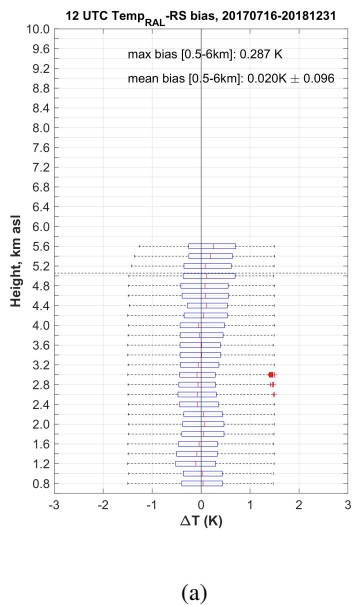

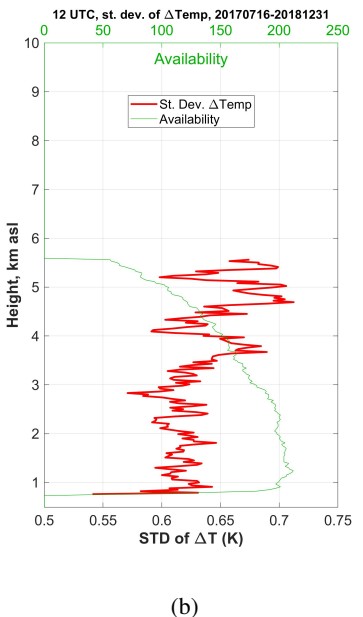

(a)             (b)

**Figure 11.** Daytime bias and STD of $\Delta T$ over the period July 2017-December 2018. In fig.11a, on each box of $200\,\mathrm{m}$ vertical span, the central mark indicates the median, and the left and right edges of the box indicate the $25^{th}$ and $75^{th}$ percentiles, respectively. The whiskers extend to the most extreme data points not considered outliers, and the outliers are plotted individually and shown by the '+' symbol. Fig.11b shows the vertical profile of standard deviation (thick red) calculated over the altitude-decreasing number of $\Delta T$ points (thin green).

hour $T_{\mathrm{ral}} - T_{\mathrm{cos}}$ differences have been collected yielding a mean daily cycle at the level $1.4\,\mathrm{km}$ to $1.7\,\mathrm{km}$ a.s.l.. The comparison in Fig. 12 shows that RALMO does not suffer any systematic daily $\Phi-$dependent bias.

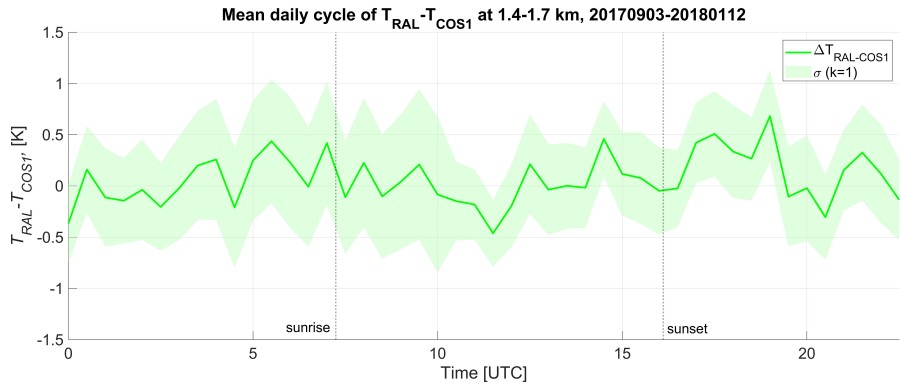

**Figure 12.** RALMO-COSMO-1 temperature at 1400-1700 m asl during Sept. 2017-Jan. 2018. The green shading accounts for the $k = 1$ standard deviation of the differences. The dashed vertical lines show the sunrise and sunset time, the mean daily cycle shows that no artefacts are introduced by the application of $f(\Phi)$ during daylight.





## 6   Seasonality study

In order to study the seasonal effects on $\mu$ and $\sigma$, the $\Delta T$ dataset has been divided into seasons. The four seasons are defined as it follows: summer, from $1^{st}$ of June to $30^{th}$ of August, autumn from $1^{st}$ September to $30^{th}$ of November); winter from $1^{st}$ of December- $15^{th}$ of March; Spring from $16^{th}$ of March to $31^{st}$ of May. Because of the less favourable conditions in winter due

to precipitation and low clouds, only few temperature profiles are available during this period. Additionally, during winter 2018, from January till mid-March, RALMO measurements have been stopped for about $80\%$ of the time due to maintenance works. The results are summarized in Tables 6 and 7 in terms of $\mu$, $\sigma$ and maximum availability $N_{max}$ over the lower tropospheric range $0\,\mathrm{km}$ to $6\,\mathrm{km}$ for daytime and over $0\,\mathrm{km}$ to $10\,\mathrm{km}$ for nighttime. With the only exception of winter, the other seasons have enough profiles to perform a statistical analysis and to draw quantitative conclusions about the contribution of each season

to the overall values $\mu$ and $\sigma$.

### 6.1   Seasonal daytime temperature statistics

The seasonal daytime $\mu$ and $\sigma$ profiles are analyzed to understand if sources of systematic errors other than $SB$ affect the retrieved $T_{\mathrm{ral}}$. The seasonal profiles are shown in Fig. (13) and (14) and summarized in Table 6. Due to the less favourable weather conditions and the maintenance works, the winter statistics count only 8 profiles. The statistical characterization of the

winter dataset can then only be qualitative. Summer and spring are the seasons with the minimum values of $\Phi$ at noon, during these two seasons $T_{\mathrm{ral}}$ is most affected by $SB$. If uncorrected for $f(\Phi)$, the noon $T_{\mathrm{ral}}$ suffers a negative mean bias of about $2\,\mathrm{K}$ at all heights (not shown here). Summer counts more than twice the number of cases in the spring dataset, nevertheless for both seasons the values of $\mu$ are compatible with a zero bias within their uncertainties (both $\sigma = 0.64\,\mathrm{K}$). Despite the less favourable weather conditions compared to spring and summer, autumn is the season with most cases and this is because there

are two autumn seasons in the dataset. Likewise spring and summer, also autumn has $\mu$ compatible with the zero-bias within its uncertainty. Through the four seasons the mean $\sigma$ spans from $0.4\,\mathrm{K}$ to $0.65\,\mathrm{K}$.

**Table 6.** Seasonal bias and $TD$ at 12 UTC

| season | $\mu$ [K] | $\sigma$ [K] | $Nmax$ |
|---|---|---|---|
| | $0.5 - 6$ km a.s.l. | | |
| summer | +0.03 | 0.64 | 74 |
| autumn | +0.005 | 0.61 | 102 |
| winter | +0.24 | 0.41 | 8 |
| spring | -0.03 | 0.64 | 31 |





Fig. (13) suggests that $T_{\mathrm{ral}}$ is not affected by any obvious systematic error ($\mu \simeq 0 \in [-\sigma, +\sigma]$) and no seasonal cycle appears in the statistics. From the perspective of the statistical validity of the studied data, any sub-sample chosen randomly from the total $T_{\mathrm{ral}}$ dataset can be described by the same $\mu$ and $\sigma$ that characterizes $\Delta T$.

Differently from the Fig. (13), the $\sigma$ profiles in Fig. (14) show different behaviors in summer-autumn and in winter-spring. The $\sigma$-profiles in Figs. (14a) and (14b) undergo a decoupling between the lower and upper part of the profile with inversion point slightly higher in autumn. An increase in $\sigma$ with height is something expected and can be explained with the decreased data availability and the decreased $SNR$ due to the increasing distance from the laser emission. However, the abrupt increase in spread at $\approx 3$ km in summer and at $\approx 4.5$ km in autumn is more related to the atmospheric dynamics than to the $SNR$. In summer, the transition between the boundary layer and the free troposphere is a region of high variability in terms of temperature and humidity. The alternating cold downdrafts and warm updrafts engendered by the overall fair weather conditions and the continuous development of thermals through the boundary layer (Martucci et al. (2010)) cause a large variability of $T_{\mathrm{ral}}$ at $\approx 3$ km, which translates into large $\sigma$-values. In autumn, the thermal activity at the top of the boundary layer is less pronounced than in summer, on the other hand, a temperature inversion linked to the formation and dissipation of stratus clouds above Payerne occurs at $\approx 4.5$ km causing larger discrepancies in the comparison with $T_{\mathrm{ors}}$.

## 6.2 Seasonal nighttime temperature statistics

At nighttime, $f(\Phi)$ has no impact on the temperature retrieval and the seasonal statistics can reveal sources of systematic error other than the $SB$ causing $|\mu| > 0$. The separation into seasons, helps understanding if the overall zero-bias shown in Fig. 10 hides seasonal non-zero biases that cancel out when combined in the full dataset. Compared to daytime cases, the availability of the nighttime dataset is higher including the one of winter cases that allows now to perform a statistical analysis. Indeed, the number of cases in the nighttime dataset is 245, versus 215 in the daytime. All seasonal $\mu$ values are compatible with the zero bias along the troposphere within $[-\sigma, +\sigma]$. Likewise the daytime seasonal statistics, also the nighttime do not reveal any obvious source of systematic error. The mean $\mu$ and $\sigma$ in the troposphere are summarized in Table 7.

**Table 7.** Seasonal bias and $TD$ at 00 UTC

| season | $\mu$ [K] | $\sigma$ [K] | $Nmax$ |
|---|---|---|---|
| | 0.5 − 6 km a.s.l. | | |
| summer | +0.11 | 0.66 | 77 |
| autumn | +0.02 | 0.65 | 118 |
| winter | +0.18 | 0.60 | 19 |
| spring | +0.08 | 0.64 | 31 |

The nighttime cases undergo different dynamics than the daytime cases. The absence of solar radiation removes almost all convection in the boundary layer and minimizes the variance of the temperature at the top of the nocturnal and residual layers.







**Figure 13.** Seasonal daytime bias of $\Delta T$ over the period July 2017-December 2018. The boxplot characteristics are the same as in Fig. 10 and Fig. 11, but restricted over the seasonal periods. Based on the definition of seasons provided in the text, panel 13a shows the summer data, panel 13b shows the autumn data, panel 13c shows the winter data, panel 13d shows the spring data.





**Figure 14.** Seasonal daytime STD of $\Delta T$ over the period July 2017-December 2018. The vertical profiles of standard deviation (thick red) are calculated over the altitude-decreasing number of $\Delta T$ points (thin green). Based on the definition of seasons provided in the text, panel 14a shows the summer data, panel 14b shows the autumn data, panel 14c shows the winter data, panel 14d shows the spring data.





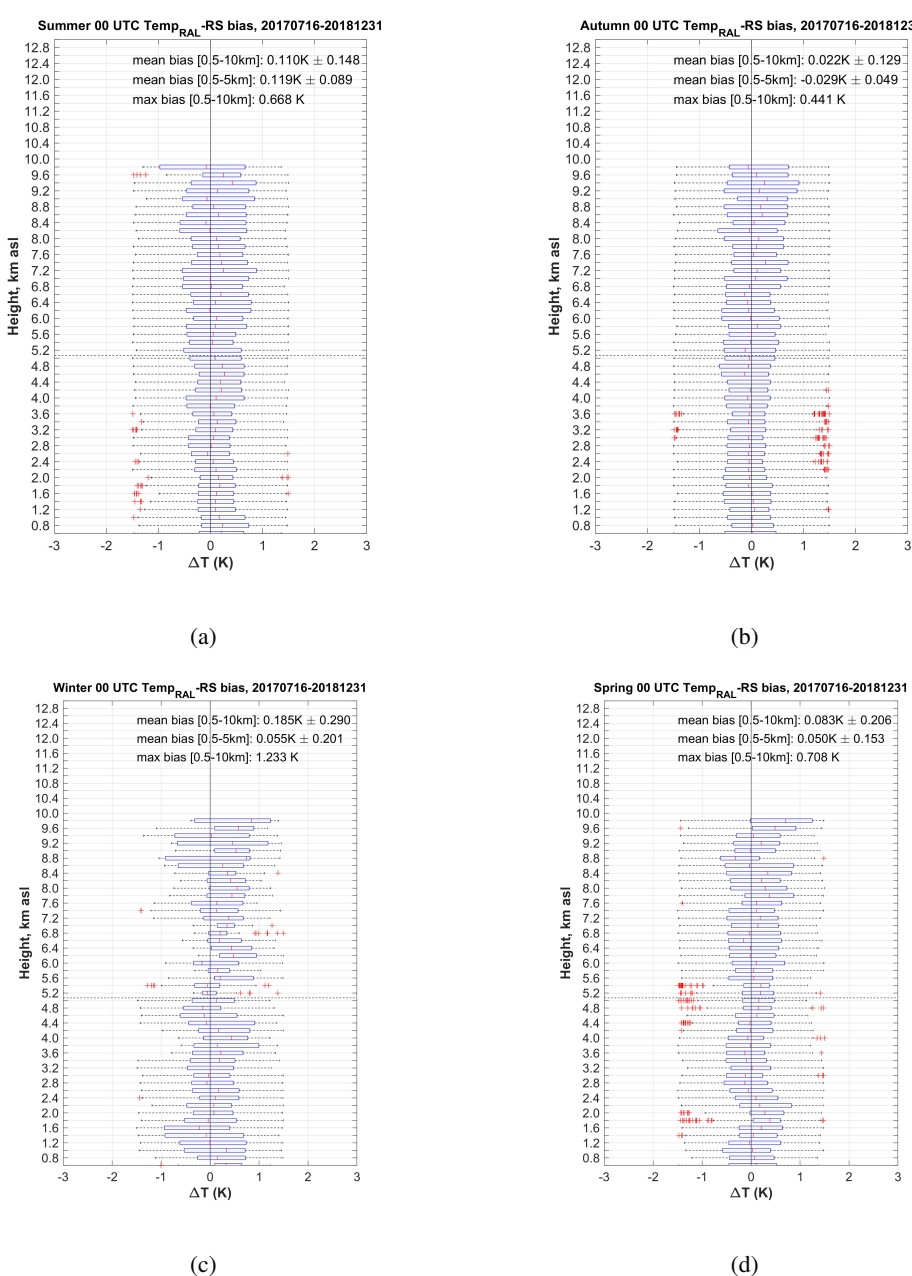

**Figure 15.** Seasonal nighttime bias of $\Delta T$ over the period July 2017-December 2018. The boxplot characteristics are the same as for Fig. 10 and Fig. 11, but restricted over the seasonal periods. Based on the definition of seasons provided in the text, panel 15a shows the summer data, panel 15b shows the autumn data, panel 15c shows the winter data, panel 15d shows the spring data.





Consequently, no sharp increase of $\sigma$ is detected at specific levels during any of the seasons. The $\sigma-$profiles increase in value with height as a response to the drop of the $SNR$ due to the distance from the emission.

# 7  Measurement of supersaturation in liquid stratus clouds

A reliable real-time measurement of the temperature in the troposphere along with a reliable measurement of the humidity
allows to calculate trustworthy profiles of relative humidity. A validation of the relative humidity measured by RALMO has been performed by Navas-Guzmán et al. (2019). The authors have characterized the relative humidity (RH) measured by RALMO using a similar procedure like for the temperature, finding that in the first 2 km the RH suffers a mean systematic and random error of $\Delta$RH $= +2\% \pm 6$ % RH.

When studying cases of cloud supersaturation a great accuracy is needed, especially when the supersaturation is assessed
at the cloud base or inside a fog layer where the supersaturation is at its onset value. Previous studies (Hudson et al. (2010); Martucci and O'Dowd (2011)) show that for different types of liquid stratus clouds forming within continental (*polluted*) or marine (*clean*) air masses, the characteristic values of supersaturation span between 0.1 % to $> 1$ %, respectively. In this sense, we cannot use here the RH measurements quantitatively, as the RH relative error is bigger than the expected maximum supersaturation. However, this limitation does not prevent to perform a qualitative study about the occurrence of supersaturation
in liquid clouds. The two case studies presented in Figures 17 and 18 show the temporal evolution of the RH and the total back-scattering ratio (BSR) for two liquid stratus clouds. The RH time series shown in the Figures 17a and 18a have maxima co-located in time and space with the maxima of the BSR (Figs. 17b and 18b), i.e. where the actual stratus clouds are located. A LIDAR cannot measure through a fully developed liquid cloud as the laser becomes totally extinct after 2-3 optical depths above the cloud base ($\approx 100-150$ m penetration). For this reason, normally, the retrieved profiles from a ground-based LIDAR
refer strictly to the lower part of the stratus cloud. The cloud cases presented here, are not completely opaque stratus and so they allow a partial LIDAR return from higher altitudes above the cloud. In Fig. 17b, the observed maxima along the RH profiles for the case of $15^{th}$ of November 2017 occur during 21:30-22:00 UTC and between 860 m and 950 m. The RH maxima during these temporal and spatial intervals reach RH$= 102.16\%$ at 21:30 UTC and RH$= 102.53\%$ at 22:00 UTC, by correcting the RH for its mean bias in the first 2 km ($\Delta$RH $= +2\%$), the resulting supersaturation is in the range $ss = 0.16 - 0.53\%$. These
values of $ss$ are typical for continental warm stratus cloud and, although qualitative, fit very well the expected microphysical scenario.

The case of the $20^{th}$ of May 2018 in Figure 18 shows the convective growth of the boundary layer above Payerne from the late morning till the central hours of the day. At the top of the developing convective boundary layer, fair-weather cumulus clouds form starting from very low altitude above the ground before 11:00 UTC and reaching $\approx 2$ km between 14:00 UTC and
15:00 UTC. The RH (Fig. 18a) reach local maxima of supersaturation correspondingly to the maxima in BSR (Fig. 18b). By performing the same qualitative analysis as for the previous case, the supersaturation is observed at 11:30 UTC between 830 m and 1160 m asl ($ss = 100.43\% - 102.32\%$) and at 14:00 UTC between 1400 and 1520 m asl ($ss = 102.25\% - 102.93\%$).



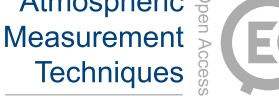

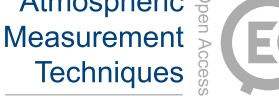

(a)

(b)

(c)

(d)

**Figure 16.** Seasonal nighttime STD of $\Delta T$ over the period July 2017-December 2018. The vertical profiles of standard deviation (thick red) are calculated over the altitude-decreasing number of $\Delta T$ points (thin green). Based on the definition of seasons provided in the text, panel 16a shows the summer data, panel 16b shows the autumn data, panel 16c shows the winter data, panel 16d shows the spring data.





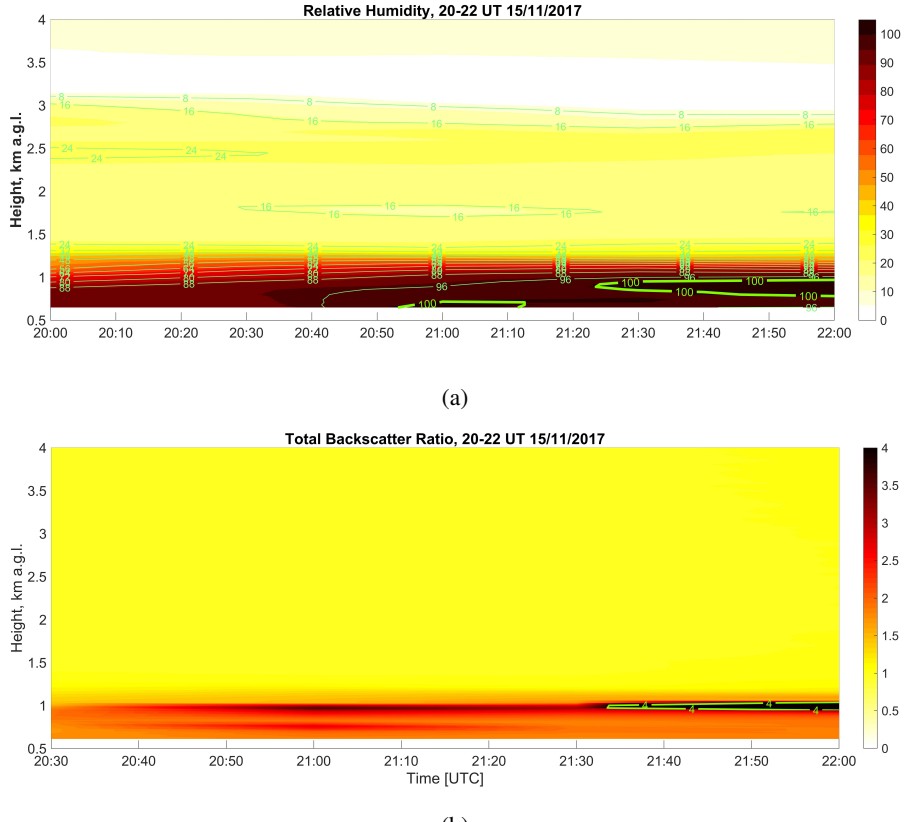

(a)

(b)

**Figure 17.** Case of 15 Nov. 2017. Top panel: RH supersaturation in a liquid stratus cloud (thick contours show the zone in the cloud where supersaturation occurs). Bottom panel: total backscatter ratio (contour lines show zones where the cloud has the largest backscatter ratio).

Correcting the RH for its mean bias in the first 2 km, the resulting supersaturation is achieved only at its high-end for the first event with $ss = 0.32\%$ and results in the range of values $ss = 0.25\% - 0.93\%$ for the second event.

## 8 Conclusions

More than 450 LIDAR temperature profiles have been compared to temperature profiles measured by the reference radiosounding system at Payerne at 11 UTC and 23 UTC during 1.5 years (July 2017–December 2018). The reference radiosounding systems (SRS-C50 and Vaisala RS41) have been validated by the GRUAN-certified Vaisala RS92 sonde in the framework of the quality assurance programme carried out at Payerne. A semi-empirical modification has been developed and applied to the background correction procedure to reduce a daytime bias. The temperature profiles retrieved from RALMO PRR data show an excellent agreement with the reference radiosounding system during both daytime and nighttime in terms of maximum bias ($\Delta T_{max}$), mean bias ($\mu$) and standard deviation ($\sigma$). The $\Delta T_{max}$, $\mu$ and $\sigma$ of the daytime differences $\Delta T = T_{ral} - T_{ors}$ over the tropospheric region $0.5-6$ km are 0.28 K, $0.02 \pm 0.1$K and $0.62 \pm 0.03$K, respectively. The nighttime $\Delta T$ dataset is character-

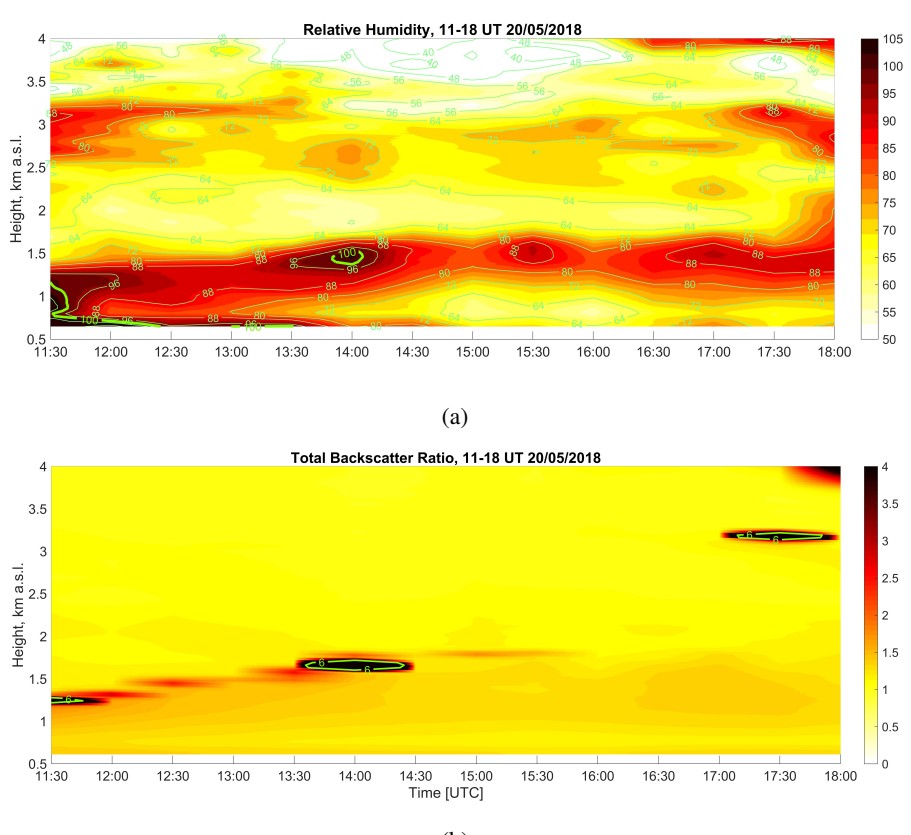

**Figure 18.** Case of 20/May/2018. Top panel: RH supersaturation in a liquid stratus cloud (thick contours show the zone in the cloud where supersaturation occurs). Bottom panel: total backscatter ratio (where the cloud has the largest backscatter ratio).

ized by a mean bias $\mu = 0.05 \pm 0.34$ K and $\sigma = 0.66 \pm 0.06$ K, while $\Delta T$ is smaller than $\Delta T_{max} = 0.29$K at all heights over the tropospheric region 0.5−10 km. We further compared the lidar data against model output and found no daytime dependence of the bias nor the standard deviation and conclude that essentially the same data quality is achieved for day and night. A season-

5    ality study has been performed to help understanding if the overall daytime and nighttime zero-bias hides seasonal non-zero biases that cancel out when combined in the full dataset. The study reveals that all independent seasonal contributions of $\mu$ are compatible with the zero-bias within their uncertainty. In general, the seasonal datasets confirm the fact that sub-sampling the total $\Delta T$ dataset, the sub-samples can still be described by the same $\mu$ and $\sigma$. The validated $T_{\text{ral}}$ has then been used to calculate the relative humidity using the humidity profiles also provided by RALMO. The relative humidity product has been validated in a parallel work by Navas-Guzmán et al. (2019) that shows that in the first 2 km the RH suffers a mean systematic

10    and random error of $\Delta$RH $= +2\% \pm 6$ % RH. The validated RH data have been used to perform a qualitative study to assess the supersaturation of water vapour in liquid stratus clouds measured by RALMO. Two cases have been investigated and the observed supersaturation values, once corrected for the RH systematic error, found compatible with the values characteristics



for continental liquid stratus clouds ($ss = 0.16 - 0.53\%$). The possibility to study supersaturation is critical to disentangle the microphysics of liquid clouds and better predict the amount of liquid water within the cloud.

We have shown that RALMO temperature profiles meet the OSCAR breakthrough uncertainty requirement of $1\,K$ for high resolution NWP (https://www.wmo-sat.info/oscar/requirements). Combined with the water vapor measuements the Raman lidar

5  has a high potential to improve NWP through data assimilation as we have demonstrated recently (Leuenberger et al., 2020) and MeteoSwiss plans to assimilate the Raman lidar in Payerne operationally in the near future.

*Acknowledgements.* This work has been supported by the Swiss National Science Foundation (project no. PZ00P2 168114).



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
