# Peer review of "Validation of temperature data from the RAmman Lidar for Meteorological Observations (RALMO) at Payerne. An application to liquid cloud supersaturation."

_Atmospheric Measurement Techniques, 2020_

## Referee Comment (RC1) · Anonymous Referee #3 · 14 Oct 2020

Summary:

The manuscript takes a comprehensive look at the temperature measurements with the Raman Lidar at the MeteoSwiss station in Payerne, RALMO. The focus is on the validation of the measurements, but a detailed description of the experimental setup is also given. RALMO utilizes the pure rotational Raman (PRR) technique, but in contrast to other instruments the PRR signals are separated by two fiber-coupled consecutive grating spectrographs. The technical design showcases an impressive long-term stability, which makes comparisons with radiosonde soundings for calibration purposes

a rare necessity. It follows a detailed description of the processing of the measurement signals from their recording, correction for dead time and background effects to the determination of the calibration constants of the temperature measurement. The error budget is also discussed. The validation of the PRR temperatures is the central part of the manuscript. For this purpose, a measurement data set of several years is used, which is compared to a large number of quality-tested temperature profiles from local radiosonde ascents or model data. Different aspects such as diurnal cycle and seasonality are investigated. The agreement is impressively good, as evidenced by various statistical parameters. Finally, as an application example, the water vapor supersaturation in liquid water clouds is examined. Besides the PRR temperatures the water vapor mixing ratio, also measured by RALMO, is used for this purpose. This study could have been more detailed, however, some questions remain open. In view of the considerable length of the manuscript and the fact that description of the lidar instrument, and data evaluation and validation of the PRR temperatures are clearly the focus of the paper, the authors should consider to remove this section from the manuscript and publish it separately. In summary, the manuscript is well written, the results are important and worth publishing. Only some polishing is recommended.

Section 1:

1. The last two paragraphs should be combined to avoid repetition.

Section 3:

1. Page7, Line 8: If only 2 of the 4 telescopes are used for temperature and humidity measurements, what are the other 2 telescopes for?

2. P7, L7: The tilted filter induces polarization effects. Have polarization issues been studied?

3. P8, L6: Please name type and manufacturer of the PMTs.

Section 4:

1. P12, L16: Probably, the step width is 0.01 ns?

2. P12, L26: The dead times differ significantly. Do you have an explanation? Do you use different PMTs?

3. P13, L4ff.: Why is this so? At 50-60 km, the 'weaker' (as you say) J_high signal should contain only background photons, and so, in theory, background subtraction should be OK.

4. P15, Fig. 9 (and others): The temperature profiles are presented starting at 500 (or 600)m. Given the fact that Payerne is at about 450 m asl, this is quite close to the ground and probably within the region of incomplete overlap. At what altitude does RALMO reach full overlap? Do you have instances where an incomplete overlap may have caused measurement errors?

Section 5:

1. P17, L1ff.: Exclusion of measurements within clouds from the statistics are justified by the attenuation of the signals and the subsequent increase in SNR. Because of the proximity of the elastic line, however, blocking might be an issue as well. Have the authors attempted to measure PRR temperatures in clouds? How well does the double-polychromator setup suppress elastic light in the PRR signals? Up to which backscatter ratio (BSR) can the PRR temperature be considered unaffected by particle scattering? Are there any polarization effects?

Section 6:

1. P25, L20: The 'clouds' presented are actually extremely thin. Even if the stratus were broken, to obtain a mean BSR of only 4 would mean that most of the integration time there was no cloud at all, or only swollen aerosols were present. Profiles of the cloud optical properties [backscatter coefficient, extinction coefficient, lidar ratio (, and depolarization ratio; but probably not available)] plus RALMO humidity and PRR temperatures would make it possible to assess the measurement situation and the

RALMO performance much better. Co-location of maximum RH and BSR sounds a bit suspicious, s. blocking comment above. As already mentioned in the summary, the reviewer recommends to discard this section.

Math, equations and running text (all):

1. All variables must be in italic.

2. If not a variable, text must not be italic, e.g.: $O_2$, $N_2$, high, low, Stokes, AntiStokes, fit, sig, SB, TD, season, max, ss, . . .

Figures:

1. Fig. 4: There is no wavelength scale as stated in the caption.

2. Fig. 5: Is there a 'degree' symbol after 'to Aerosol &T'?

3. Fig. 5: The depiction of the water vapor spectrum would be more realistic if the steep slope was on the blue shoulder.

4. Fig. 6: There are many more holes in the blocks (at the edges) than explained in the running text. What are they for?

5. Figs. 10, 11: Use same style for panels left and right. Use same x range for STD in both figures.

6. Figs. 14, 16: Harmonize x ranges as much as possible. For instance, use 0-120 for availability in all panels, 0-1 for STD.

Tables:

1. Tab. 2: There are entries missing down in the third column.

Typos:

1. P3, L5: '. Our'

2. P3, L13: '. Moreover,'

3. P3, L19: 'possible causes'

4. P4, L7: '2018)'

5. Caption Fig. 2: '2b).'

6. Caption Fig. 3: 'figures.'

7. P6, L30: 'transceiver'

8. P7, L2: 'of the signal'

9. Caption Fig. 6: This is not the correct text (has been copied from Fig. 5).

10. P11, L11: 'is used to'

11. P17, L16: 'are the metric'

12. Caption Fig. 10: Explain 'STD'.

13. Caption Fig. 11: Explain 'STD'.

14. Caption Fig. 12: 'Differences between RALMO and COSMO temperatures'

15. Caption Fig. 12: Include date of sunrise and sunset plotted.

16. P20, L2: 'November;'

17. P20,L20: 'Like spring'

18. Heading, Tab. 6: Explain 'TD'.

19. P21, L8: 'from the instrument'

20. Heading, Tab. 7: Explain 'TD'.

21. P25, L24: Define 'ss'.

---

## Referee Comment (RC2) · Anonymous Referee #1 · 6 Nov 2020

The authors describe a study for the validation of temperature data from the RAman Lidar for Meteorological Observations (RALMO) at Payerne (July 2017 to December 2018) against two reference operational radiosounding systems (ORS) co-located with RALMO. The overall quality of the preprint, is characterized by well-illustrated methods and scientific significance, contributing to scientific progress within the scope of Atmospheric Measurement Techniques. GENERAL COMMENTS There are too many numbers presented on the abstract and it is hard to understand the meaning of each one. Although it would be easier to pull out the main message if an additional table showed

all the values separately for daytime and nighttime measurements, declaring also that daytime corresponds to lower troposphere and nighttime to whole troposphere. SPE-CIFIC COMMENTS During daytime measurements in the lower troposphere, and nighttime measurements in whole troposphere, mean bias are found $\mu$ = 0.02 $\pm$ 0.1 K and $\mu$ = 0.05 $\pm$ 0.34 K respectively. How would you comment those big errors? TECHNICAL CORRECTIONS Page 1 Line 18-19 "imperfect subtraction" of the background from the daytime PRR profiles induces a bias of up to 2 K at all heights. In which figure is this represented? Page 3 Line 3 "Trustworthy references can be provided by co-located radiosondes, satellites or a numerical models." Propably "a" should be removed. Figure 1 (and for all similar graphs) St. dev and median are calculated after averaging, so maybe it should be mentioned at the label. Page 4 Line 9-10 "Figures 1 and 2 show the statistical biases of the SRS-C50 and the RS41 with respect to the reference RS92 as a function of height for the day and night-time launches." In order to be better-structured, figures or the SRS-C50, RS41 should be reversed. Page 11 Line 23 "Once the signals corrected, their ratio is used used to retrieve the temperature" the word used should be removed.

---

## Author Comment (AC1) · 1 Dec 2020

[a4paper,14pt]extarticle

etoolbox mathptmx [11pt]moresize blindtext, xfrac xcolor geometry   a4paper, total=170mm,257mm, left=20mm, top=20mm,  graphicx caption

[Figure]

**1   Answer to Referee's general comment**

We thank the Referee for the detailed and constructive review provided. We can see that our article is improved by implementing your suggestions.

We appreciate your suggestion to remove the part on cloud supersaturation and to rather put it in a separated publication. We understand that the paper is already long and, even without the supersaturation study, presents already all useful results needed to validate thoroughly the temperature product. Even if the application to a real case of liquid cloud supersaturation is important and provides a direct tool to validate a product that is largely undersampled currently across the scientific community, we agree to remove the chapter on supersaturation. This part will be integrated in a separate publication and submitted as new scientific article.

**2   Answer to Referee's specific comments**

Sect.1, The last two paragraphs should be combined to avoid repetition.
Done.
Sect.3, Page7, Line 8: If only 2 of the 4 telescopes are used for temperature and humidity measurements, what are the other 2 telescopes for?
all four telescopes collect the backscattered signal and transmit it through the REF to the $H_2O$ polychromator. Two mirrors reflects the collected backscattered signal onto the REF and transmit it to the temperature polychromator.
Sect.3, P7, L7: The tilted filter induces polarization effects. Have polarization issues been studied?
Yes. There is a full study that has been conducted, because we are in the process to implement a new depolarization channel. The REF at $9°$ induces only a negligible depolarisation $s$. The figure 1 below shows that all signals $(p + s)$ are reflected by the REF without modification.

Sect.3, P8, L6: Please name type and manufacturer of the PMTs.
Done.
Sect.4, P12, L16: Probably, the step width is 0.01 ns?
Indeed!! Corrected, thanks!
Sect.4, P12, L26: The dead times differ signifiAcantly. Do you have an explanation?
Do you use different PMTs?
Yes each channel has a a dedicated PMT. In principle, the dead-time is provided as
a specification of the acquisition card (in this case the FastCom P7888), but the PMT,
from which the acquisition card gets the number of photons can modify the deadtime
significantly.
Sect. 4, P13, L4ff.: Why is this so? At 50-60 km, the 'weaker' (as you say) $J_{high}$ signal
should contain only background photons, and so, in theory, background subtraction
should be OK.
Yes, "it should", and that is why the standard procedure takes the far range averaged
signal and subtract it from the entire signal. However, from a computational point of
view, the operation of calculating a mean value (MATLAB in our case) introduces a very
small error, due to the not limitless precision of this operation. A slight overestimation
or underestimation with respect to the true value generates fractions of percentage-
error after subtraction that then cause the temperature to have up to 1 K or 2 K bias.
Sect. 4, P15, Fig. 9 (and others): The temperature profiAles are presented starting at
500 (or 600)m. Given the fact that Payerne is at about 450 m asl, this is quite close to
the ground and probably within the region of incomplete overlap. At what altitude does
RALMO reach full overlap? Do you have instances where an incomplete overlap may
have caused measurement errors?
Payerne station is at 491 m a.s.l., RALMO's full overlap occurs at an altitude of
$\approx$ 4000 m. However, the temperature measurement is not really affected by the in-
complete overlap in the region 500-4000 m a.s.l. as it is proportional to the ratio
$J_{low}(z)/J_{high}(z)$ and both $J_{low}(z)$ and $J_{high}(z)$ have the same overlap function.

REF_depol.jpg

**Fig. 1.** Reflectance for two tilt angles (angle of incidence, AOI) of $9°$ and $20°$; the half-angle cone of the reflected light is $8.5°$. The data are computed by the manufacturer's on-line simulation tool

Sect. 5, P17, L1ff.: Exclusion of measurements within clouds from the statistics are justified by the attenuation of the signals and the subsequent increase in SNR. Because of the proximity of the elastic line, however, blocking might be an issue as well. Have the authors attempted to measure PRR temperatures in clouds? How well does the double-polychromator setup suppress elastic light in the PRR signals? Up to which backscatter ratio (BSR) can the PRR temperature be considered unaffected by particle scattering? Are there any polarization effects? "

This is a very interesting point. The double-stage temperature polychromator has an almost $100\%$ efficiency in removing the stray light from the elastic signal. However the extinction coefficient of the PRR signal backscattered from a cloudy volume is so high to impede the transmission of the PRR signal through the cloud and back to the telescope as soon as the cloudy volume is one optical depth above the cloud base $(BSR \approx 5)$.

Sect. 6, P25, L20: The 'clouds' presented are actually extremely thin. Even if the stratus were broken, to obtain a mean BSR of only 4 would mean that most of the integration time there was no cloud at all, or only swollen aerosols were present. Profiles of the cloud optical properties [backscatter coefficient, extinction coefficient, lidar ratio (, and depolarization ratio; but probably not available)] plus RALMO humidity and PRR temperatures would make it possible to assess the measurement situation and the RALMO performance much better. Co-location of maximum RH and BSR sounds a bit suspicious, s. blocking comment above. As already mentioned in the summary, the reviewer recommends to discard this section.

The studied clouds (both cases) are indeed thin liquid stratus (15 Nov 2017) and fair weather cumulus (20 May 2018) clouds. In both cases, the colocated ceilometer detected a cloud base. An opaque cloud detected by the ceilometer starts at about BSR>2. Both cases showed:

1. Pre-cloud formation, with fast growing higroscopicity as a precursor of cloud.

2. Already formed cloud.

3. blocking by fully-developed cloud

The second stage is the one when supersaturation has its onset at the cloud base. All profiles (backscatter, extinction, humidity...) are available and could be added as ancillary information. However, we have decided as mentioned in the general comment to drop the section on supersaturation as suggested by the Referee. This part will hopefully converge into a cloud microphysics dedicated paper.

**Math, equations and running text (all)**:

1. All variables must be in italic. Done

2. If not a variable, text must not be italic,e.g.: $O_2$,$N_2$,high,low,Stokes,AntiStokes, fit, sig, SB, TD, season, max, ss, ... Done, apart for $ss$ that is a variable.

**Figures**:

1. Fig. 4: There is no wavelength scale as stated in the caption.
corrected

2. Fig. 5: Is there a 'degree' symbol after 'to Aerosol & T'?
Yes.

3. Fig. 5: The depiction of the water vapor spectrum would be more realistic if the steep slope was on the blue shoulder.
Thanks for noticing it, we have corrected the WV spectrum.

4. Fig. 6: There are many more holes in the blocks (at the edges) than explained in the running text. What are they for?
   The holes surrounding the fiber's block are not active, are available holders for other potential input/output cables. A full description has been added to the figure's caption.

5. Figs. 10, 11: Use same style for panels left and right. Use same x range for STD in both figures.
   Axis $x - y$ of the panels in figures 10-11 have been harmonized.

6. Figs. 14, 16: Harmonize x ranges as much as possible. For instance, use 0-120 for availability in all panels, 0-1 for STD.
   Done

**Tables**:

1. Tab. 2: There are entries missing down in the third column.
   Thanks for noticing it! We have added the efficiencies for lines 9 and 10

**Typos**:

1. P3, L5: '. Our'
   done

2. P3, L13: '. Moreover,'
   done

3. P3, L19: 'possible causes'
   done

4. P4, L7: '2018)
   done

5. Caption Fig. 2: '2b).
   done

6. Caption Fig. 3: 'figures.'
   done

7. P6, L30: 'transceiver'
   done

8. P7, L2: 'of the signal'
   done

9. Caption Fig. 6: This is not the correct text (has been copied from Fig. 5).
   the caption has been properly adapted.

10. P11, L11: 'is used to'
    done

11. P17, L16: 'are the metric'
done

12. Caption Fig. 10: Explain 'STD'.
done

13. Caption Fig. 11: Explain 'STD'.
done

14. Caption Fig. 12: 'Differences between RALMO and COSMO temperatures'
done

15. Caption Fig. 12: Include date of sunrise and sunset plotted.
done

16. P20, L2: 'November;'
done

17. P20,L20: 'Like spring'
done

18. Heading, Tab. 6: Explain 'TD'.
done

19. P21, L8: 'from the instrument'
    We replaced with, "from the lidar's telescope"

20. Heading, Tab. 7: Explain 'TD'.
    done

21. P25, L24: Define 'ss'.
    The chapter has been removed.

---

## Author Comment (AC2) · 1 Dec 2020

**1 Answer to Referee's general comment**

We thank the Referee for the detailed and constructive review, it helped us improving the readability and quality of our study.
We agree that the abstract brings many detailed quantitative information and that it is not easy to retain all of them. We have rephrased the part in the abstract where the statistics are provided, and therefore improved the readability. Unfortunately, it is not recommended by the AMT guidelines to include a table in the abstract. However, Table 5 brings exactly what requested by Referee#1 including the information about the portion of troposphere covered by the daytime and nighttime statistics.

**2 Answer to Referee's specific comments**

During daytime measurements in the lower troposphere, and nighttime measurements in whole troposphere, mean bias are found $\mu = 0.02 \pm 0.1$ K and $\mu = 0.05 \pm 0.34$ K respectively. How would you comment those big errors?.
The daytime $\mu = 0.02 \pm 0.1$ K and the nighttime $\mu = 0.05 \pm 0.34$ K are the mean biases along the atmospheric regions where the daytime and nighttime temperature profiles are calculated. Each mean bias is provided with a variability, i.e. the standard deviation of the biases. Based on values of $\sigma$ of 0.62 K (day) and 0.66 K (night), representing the standard deviation of all differences $\Delta T = T_{\mathrm{ral}} - T_{\mathrm{ors}}$ over which $\mu$ is calculated, a variability of 0.1 K (day) and 0.34 K (night) can be expected.

**3 Answer to Referee's Technical corrections**

:

1. Page 1 Line 18-19 "imperfect subtraction" of the background from the daytime PRR profiles induces a bias of up to 2 K at all heights. In which figure is this represented?
   The results have not been included in the paper because we deemed that it was out of the scope of the study. However, the sensitivity test of the calculated temperature with respect to a variation of 1% of the ratio $J_{low}/J_{high}$ has been carried out and is shown in figure 1. The plot in panel (b) shows as a change of 1% can induce up to 2 K difference in the lower part of the profile.

2. Page 3 Line 3 "Trustworthy references can be provided by co-located radiosondes, satellites or a numerical models." Probably "a" should be removed.
   done

[Figure]

[Figure]

(a)                                      (b)

Figure 1: Differences in the calculated temperature induced by a relative change of 1% in the ratio $J_{low}/J_{high}$.

3. Figure 1 (and for all similar graphs) St. dev and median are calculated after averaging, so maybe it should be mentioned at the label.
   The median and St. Dev. have been removed from Figures 1-3 after the technical review.

4. Page 4 Line 9-10 "Figures 1 and 2 show the statistical biases of the SRS-C50 and the RS41 with respect to the reference RS92 as a function of height for the day and night-time launches." In order to be better-structured, figures or the SRS-C50, RS41 should be reversed.
   Done

5. Page 11 Line 23 "Once the signals corrected, their ratio is used used to retrieve the temperature" the word used should be removed.
   Done.